# A proximity biotinylation-based approach to identify protein-E3 ligase interactions induced by PROTACs and molecular glues

Satoshi Yamanaka [1], Yuto Horiuchi[1], Saya Matsuoka[1], Kohki Kido[1], Kohei Nishino[2], Mayaka Maeno[3], Norio Shibata [3], Hidetaka Kosako[2] & Tatsuya Sawasaki [1✉]

Proteolysis-targeting chimaeras (PROTACs) as well as molecular glues such as immuno-modulatory drugs (IMiDs) and indisulam are drugs that induce interactions between substrate proteins and an E3 ubiquitin ligases for targeted protein degradation. Here, we develop a workflow based on proximity-dependent biotinylation by AirID to identify drug-induced neo-substrates of the E3 ligase cereblon (CRBN). Using AirID-CRBN, we detect IMiD-dependent biotinylation of CRBN neo-substrates in vitro and identify biotinylated peptides of well-known neo-substrates by mass spectrometry with high specificity and selectivity. Additional analyses reveal ZMYM2 and ZMYM2-FGFR1 fusion protein—responsible for the 8p11 syndrome involved in acute myeloid leukaemia—as CRBN neo-substrates. Furthermore, AirID-DCAF15 and AirID-CRBN biotinylate neo-substrates targeted by indisulam and PRO-TACs, respectively, suggesting that this approach has the potential to serve as a general strategy for characterizing drug-inducible protein–protein interactions in cells.

[1] Division of Cell-Free Sciences, Proteo-Science Center, Ehime University, Matsuyama 790-8577, Japan. [2] Division of Cell Signaling, Fujii Memorial Institute of Medical Sciences, Tokushima University, Tokushima 770-8503, Japan. [3] Department of Nanopharmaceutical Sciences, Nagoya Institute of Technology, Nagoya 466-8555, Japan. ✉email: sawasaki@ehime-u.ac.jp

Thalidomide was initially used as a sedative[1,2], but it has been reported to have severe teratogenic effects[1,2]. Currently, thalidomide and its derivatives (immunomodulatory drugs; IMiDs), including lenalidomide, pomalidomide, and 5-hydroxythalidomide (5HT, Fig. 1a), have been identified; lenalidomide and pomalidomide are the main drugs used for treating certain hematologic cancers[3,4]. IMiDs directly interact with cereblon (CRBN)[5], recruits a key substrate (called a neo-substrate) for CRL4$^{CRBN}$, and subsequently facilitates degradation of neo-substrates by the 26S proteasome[6–11]. For example, IMiD-dependent IKZF1 and IKZF3 protein degradations are involved in

the anti-MM effect[6,7], and SALL4 (spalt-like transcription factor 4) and PLZF (promyelocytic leukaemia zinc finger protein) degradations are involved in teratogenicity[9–11]. Since the main role of IMiDs is to facilitate interaction between CRBN and neo-substrate proteins[6–11], analysis of IMiDs-inducible protein–protein interactions (PPIs) is crucial for understanding the biological roles of IMiDs.

A small molecule in cells can regulate biological functions via the induction of PPIs[5–12]. For instance, major phytohormones such as auxin, abscisic acid (ABA), gibberellic acid (GA), and jasmonic acid (JA) induce PPIs as the first key event in plant

**Fig. 1 AirID-CRBN biotinylates neo-substrates in the presence of thalidomide and IMiDs in cells. a** Chemical structures of IMiDs used in this study. **b** Schematic diagram of IMiDs-dependent neo-substrate biotinylation by AirID-CRBN. **c** IMiDs-dependent biotinylation assay of exogenous neo-substrates by AirID-CRBN in cells. HEK293T cells stably expressing AGIA-AirID-CRBN-WT or -YW/AA were transfected with pCDNA3.1-Myc-IKZF1 and -SALL4, and treated with DMSO or 10 μM pomalidomide (Po), 5 μM biotin and 10 μM MG132 for 6 h. The experiment was repeated three times independently with similar results. **d** LC-MS/MS analysis of biotinylated peptides or proteins using three enrichment methods. MM1.S cells stably expressing AGIA-AirID-CRBN-WT were treated with DMSO or 10 μM pomalidomide (Po) in the presence of 10 μM biotin and 5 μM MG132 for 8 h (biological replicates; $n = 3$). Biotinylated peptides were enriched with tamavidin 2-REV or anti-biotin antibody, and biotinylated proteins were enriched with streptavidin. Significant changes in the volcano plots were calculated by Student's two-sided $t$-test and the false discovery rate (FDR)-adjusted $P$-values calculated using Benjamini–Hochberg method are shown in the Supplementary Data 2–4. **e** Neo-substrate selectivity of IMiDs for biotinylation by AirID-CRBN. MM1.S cells stably expressing AGIA-AirID-CRBN-WT were treated with DMSO, 20 μM thalidomide (Th), 10 μM lenalidomide (Le), 10 μM pomalidomide (Po) or 20 μM 5-hydroxythalidomide (5HT) in the presence of 10 μM biotin and 5 μM MG132 for 8 h. The experiment was repeated twice independently with similar results. **f** LC-MS/MS analysis of biotinylated peptides using AirID-CRBN. MM1.S cells stably expressing AGIA-AirID-CRBN-WT were treated with DMSO, 20 μM thalidomide (Th), 10 μM lenalidomide (Le), 10 μM pomalidomide (Po) or 20 μM 5-hydroxythalidomide (5HT) in the presence of 10 μM biotin and 5 μM MG132 for 8 h (biological replicates; $n = 3$). Biotinylated peptides were enriched with tamavidin 2-REV and analysed by LC-MS/MS. **g** Heat map of biotinylated peptides of IKZF1 or IKZF3 (IMiD/DMSO ratio >5 and $P$-value < 0.05) among IMiDs detected in LC-MS/MS analysis using MM1.S cells. **f, g** Significant changes in the volcano plots and heat map were calculated by Student's two-sided $t$-test and the false discovery rate (FDR)-adjusted $P$-values calculated using Benjamini–Hochberg method are shown in the Supplementary Data 5. **c, e** Biotinylated proteins were pulled down using streptavidin beads and analysed by immunoblotting. Source data are provided as a Source data file.

hormone responses[13]. Other drugs function similarly to IMiDs; for example, indisulam induces PPI between DCAF15 and RBM39 to confer anticancer effects on several human cell types[14,15]. Furthermore, proteolysis-targeting chimaeras (PRO-TACs), which are bifunctional molecules comprising an E3 binding compound (E3 binder) and a target binding compound (target binder)[16,17], induce PPI for target protein degradation, thus showing promise as a next-generation drug. Therefore, analysis of small molecule- or drug-inducible interacting proteins is essential for understanding the biological roles of these molecules and their binding to E3 ligase proteins.

BioID is a non-toxic labelling systems based on the biotin ligase BirA that has been widely used in diverse living samples, including cultured cells[18], mouse[19], and yeast[20]. We previously developed ancestral BirA for proximity-dependent biotin identification (AirID) and showed that it has both high biotinylating activity and proximity dependency[21]. Furthermore, we recently reported that biotinylated peptides can be efficiently detected in cells using tamavidin 2-REV[21,22], an engineered avidin-like protein with reversible biotin-binding capability[23,24]. These facts motivated us to develop an analysis system to detect drug-inducible PPIs using a combination of AirID and tamavidin 2-REV.

Here, we established a method to analyse PPIs driven by molecular glues and PROTACs by AirID and mass spectrometry to directly detect biotinylated peptides. In streptavidin pull-down assays (STA-PDAs) both in vitro and with lysates of cultured cells, well-known neo-substrates were biotinylated by AirID-CRBN or -DCAF15 in an IMiDs- or indisulam-dependent manner, respectively. Importantly, mass spectrometry analysis of biotinylated peptides from lysates of cells expressing AirID-CRBN revealed that ZMYM2 is a pomalidomide-dependent neo-substrate of CRBN. Furthermore, pomalidomide-induced protein degradation of ZMYM2-FGFR1 attenuated the STAT1/3 and ERK signalling pathways, which are reportedly involved in hematologic cancer. Finally, we showed that this method can be applied to analysis of the indisulam and PROTACs substrates. These results strongly suggest that our approach is useful for analysing drug-inducible PPIs in cells.

## Results

**AirID-CRBN can biotinylate endogenous neo-substrates in the presence of IMiDs in cells.** As biotin molecules bind to avidin/avidin-like proteins with high specificity and affinity ($K_d \sim 10^{-15}$ M), proximity biotinylation is a powerful approach for identifying interactors with a protein of interest[25]. Therefore, we attempted to establish a biotin-labelling method for detecting the interaction between CRBN and neo-substrates induced by IMiDs. As shown in Fig. 1b, we recently reported that AirID-fused CRBN (AirID-CRBN) can IMiDs-dependently biotinylate IKZF1 and SALL4, which are well-known neo-substrates of IMiDs[21]. Consistent with this report, STA-PDA showed that FLAG-GST-SALL4 and -IKZF1 were biotinylated in the presence of IMiDs in vitro (Supplementary Fig. 1a). In addition, overexpressed Myc-SALL4 was pomalidomide-dependently biotinylated by overexpressed AGIA-AirID-CRBN-WT but not by AGIA-AirID-CRBN-YW/AA (IMiDs unbound mutant[5]) in CRBN$^{-/-}$ HEK293T ells[26,27] (Supplementary Fig. 1b). Next, we generated reconstituted cell lines stably expressing AGIA-AirID-CRBN-WT (wild type) or -YW/AA by infection of CRBN$^{-/-}$ HEK293T cells with lentivirus, and we performed STA-PDA using the reconstituted HEK293T cells transiently transfected with Myc-SALL4 and Myc-IKZF1 expression vectors. Immunoblot analysis showed that AGIA-AirID-CRBN-WT biotinylated both Myc-SALL4 and Myc-IKZF1 in the presence of pomalidomide, but AGIA-AirID-

CRBN-YW/AA did not (Fig. 1c). To investigate whether AGIA-AirID-CRBN can induce biotinylation of endogenous neo-substrates, we generated THP-1 (acute monocytic leukaemia) cells stably expressing AGIA-AirID-CRBN. As a result of STA-PDA, pomalidomide-dependent biotinylation of IKZF1 and IKZF3 was detected by immunoblot analysis (Supplementary Fig. 1c).

We then evaluated the optimal conditions for detecting IMiDs-dependent interactions between CRBN and neo-substrates (Fig. 1a). First, we investigated whether additional supplementation with biotin enhanced the biotinylation of the neo-substrate. Immunoblot analysis showed that 5 or 50 μM biotin supplementation did not affect IKZF1 or IKZF3 biotinylation levels in THP-1 cells (Supplementary Fig. 1d). By contrast, IMiDs-dependent biotinylation of PLZF was increased by biotin supplementation (5 or 50 μM) in HEK293T cells (Supplementary Fig. 1e). However, in the case of the highest biotin concentration (500 μM), biotinylated PLZF was not detected (Supplementary Fig. 1e). In addition, none of the biotinylated proteins in the 500 μM lane were detected by the anti-biotin antibody (Supplementary Fig. 1e), suggesting that excessive biotin supplementation competitively inhibit the capture process of biotinylated proteins by streptavidin beads in certain cells such as HEK293T. By contrast, IMiDs concentration did not affect PLZF biotinylation in HEK293T cells (Supplementary Fig. 1f). Next, we examined whether the proteasome inhibitor MG132 was necessary to detect neo-substrate biotinylation. In CRBN$^{-/-}$ HEK293T cells stably expressing AGIA-AirID-CRBN, immunoblot analysis showed that extended incubation (24 h) with IMiDs induced protein degradation of neo-substrates, though brief incubation (8 h) did not (Supplementary Fig. 1g). On the other hand, in HuH7 cells stably expressing AGIA-AirID-CRBN in addition to endogenous CRBN, 8 h of incubation with pomalidomide induced remarkably protein degradation of neo-substrates (Supplementary Fig. 1h). These results indicate that CRL4$^{AirID-CRBN}$ interacts with neo-substrates in the presence of IMiDs, but its degradation activity was weaker than that of CRL4$^{CRBN}$. Taken together, the results showed that supplementation with 5–50 μM biotin and MG132 was expected to be optimal for detecting biotinylated neo-substrates.

LC-MS/MS analysis following enrichment with tamavidin 2-REV can directly detect peptides biotinylated by AirID[21]. Importantly, tamavidin 2-REV reversibly binds biotinylated peptides, which can be competitively eluted by excess biotin[22]. These features are advantageous for identifying interacting proteins because biotinylated sites are also identified; accordingly, tamavidin 2-REV would be expected to reduce undesired contaminations of non-biotinylated proteins from cell lysates. Therefore, we performed a pilot assay for LC-MS/MS analysis of biotinylated peptides in THP-1 cells using the above conditions. Biotinylated peptides from IKZF1 and IKZF3 were increased by treatment with thalidomide or pomalidomide (Supplementary Fig. 2, Supplementary Data 1). We then generated a multiple-myeloma cell-line, MM1.S, stably expressing AGIA-AirID-CRBN and compared the enrichment of biotinylated proteins or peptides across two conventional methods (streptavidin or anti-biotin antibody[28,29]) and tamavidin 2-REV. LC-MS/MS analyses revealed that among the three enrichment methods tamavidin 2-REV enrichment enabled us to detect the most biotinylated peptides of neo-substrates such as IKZF1, IKZF3, and ZFP91 (Fig. 1d, Supplementary Data 2–4). These results suggest that the combination of AirID-CRBN and tamavidin 2-REV is a powerful method for identifying IMiDs-dependent interactors in CRBN.

IMiDs (Fig. 1a) are known to exhibit neo-substrate selectivity. For example, casein kinase 1 alpha (CK1α) is degraded by lenalidomide, but not thalidomide and pomalidomide[8]. Furthermore, we have reported that 5HT, a primary metabolite

of thalidomide, cannot induce IKZF1 and IKZF3 protein degradation[11,26,30]. Based on these findings, we investigated whether biotinylation of AirID-CRBN provides neo-substrate selectivity for each IMiDs treatment. STA-PDA showed that 5HT scarcely induced biotinylation of IKZF1 and IKZF3 by AirID-CRBN in MM1.S cells (Fig. 1e). Consistent with STA-PDA and previous studies[11,26,30], LC-MS/MS analysis of biotinylated peptides showed that lenalidomide and pomalidomide strongly induced IKZF1 and IKZF3 biotinylations, but thalidomide and 5HT scarcely induced the biotinylations (Fig. 1f, Supplementary Data 5). The heatmap analysis extracted from the MS data clearly showed drug selectivity against IKZF1 and IKZF3 (Fig. 1g). The numbers of biotinylated peptides from pomalidomide and lenalidomide treatments were much higher than those from thalidomide and 5HT treatments (Supplementary Fig. 3a). These results indicate that AirID-CRBN and MS analysis of biotinylated peptides can be used to evaluate neo-substrate selectivity. Moreover, MM1.S cells more strongly induced IKZF1 and IKZF3 biotinylations than THP-1 cells (Supplementary Fig. 3b), indicating that this method can detect the type and quantity of biotinylated proteins among cell lines.

**AirID-CRBN with IMiDs leads to highly selective biotinylation of neo-substrates in cells**. After 2014, at least eight proteins were reported as neo-substrates that interact directly with CRBN in the presence of IMiDs and were degraded with the full-length protein form[6–11,31]. Based on STA-PDA and LC-MS/MS analysis using THP-1 and MM1.S cells, AirID-CRBN biotinylated the well-known neo-substrates IKZF1 and IKZF3 (Fig. 1d–g and Supplementary Figs. 1c, d, 2, 3). Conversely, other neo-substrates were not detected in THP-1 and MM1.S cells. Using the optimal conditions described above, we analysed whether IMiDs-inducible interactions between AirID-CRBN and known neo-substrates could be selectively detected in other cell lines (HEK293T, HuH7, and IMR32) with the four IMiDs (Fig. 1a). In HEK293T cells, the neo-substrates GSPT1, WIZ, and CK1α were detected (Fig. 2a, Supplementary Data 6). GSPT1 biotinylation was detected by treatment with all IMiDs; interestingly, biotinylation was most strongly induced by 5HT (Fig. 2b). This 5HT-inducible biotinylation was confirmed by immunoblotting (Fig. 2c). 5HT also induced GSPT1 degradation (Fig. 2d), although it was weaker than that induced by CC-885, as reported previously[31]. Furthermore, WIZ is reportedly a neo-substrate that is degraded in a pomalidomide-dependent manner[10]. Similar to this report, WIZ biotinylation was induced by pomalidomide, as detected by MS analysis (Fig. 2a) and STA-PDA (Fig. 2c). Furthermore, the biotinylated peptide of CK1α was most strongly detected by lenalidomide treatment (Fig. 2a). These biotinylation results from AirID-CRBN are consistent with a report that CK1α degradation is more strongly induced by lenalidomide than other compounds (Fig. 2d)[32]. These results indicate that IMiDs-dependent biotinylation of neo-substrate by AirID-CRBN reflects the selective interaction of three neo-substrates.

Additionally, to further investigate drug-inducible biotinylation in diverse cell lines, we selected two stable cell lines expressing AirID-CRBN: (1) HuH7 (hepatocellular carcinoma) cells because it expresses SALL4, which is a known neo-substrate involved in thalidomide teratogenicity;[9–11] and (2) IMR32 (neuroblastoma) cells because CRBN also functions in neuronal cells[33]. Immunoblot analyses after STA-PDA showed that pomalidomide treatment induced IMiDs-inducible biotinylation of known neo-substrates, such as SALL4, PLZF, and WIZ, in these cells (Supplementary Fig. 4a, b). As expected, LC-MS/MS analysis detected the biotinylation of SALL4, WIZ, and CK1α in HuH7 cells or WIZ, CK1α, and ZNF827 in IMR32 cells (Fig. 3a, b,

Supplementary Data 7, 8). Importantly, biotinylated peptides of SALL4 were the most strongly detected in the presence of 5HT in HuH7 cells (Fig. 3a, c), and this neo-substrate selectivity of 5HT is consistent with previous studies[11,26,30]. Furthermore, similar to HEK293T cells, the biotinylated peptide of CK1α was most strongly found in the presence of lenalidomide in HuH7 and IMR32 cells (Fig. 3a–d). In all cell lines, more biotinylated peptides were detected in the presence of pomalidomide or lenalidomide than in the presence of thalidomide (Supplementary Fig. 5a–c). Taken together, these results show that AirID-CRBN leads to in cell biotinylation of known neo-substrates with high selectivity and specificity for IMiDs.

From MS data of HEK293T, HuH7, and IMR32 cells, we compared the increase in biotinylated peptides following IMiDs treatment. Biotinylated peptides from HuH7 and IMR32 cells were much higher than those from HEK293T cells (Supplementary Fig. 6a). In fact, immunoblot analysis revealed that biotinylated proteins were more abundant in HuH7 and IMR32 cells than in MM1.S and HEK293T cells (Supplementary Fig. 6b). In addition, 5HT-inducible biotinylated peptides were more abundant in IMR32 cells than in other cell lines (Supplementary Fig. 5a–c). We then compared the protein-expression levels of ZNF687 (all cell lines) and ZMYM2 (IMR32 cells), which were detected as IMiDs-dependent CRBN interactors by MS analyses (Figs. 2a, b, 3a–d). As a result, the expression of ZNF687 was almost the same in all cell lines (Supplementary Fig. 6c), but that of ZMYM2 was the highest in IMR32 cells among all cell lines (Supplementary Fig. 6c). These results suggest that the type and quantity of biotinylated proteins are dependent on both types of supplemented IMiDs and protein-expression levels in each cell line.

Next, we confirmed whether the constituent proteins of CRL4[CRBN], such as DDB1 and CUL4A, or proteins interacting with CRBN, such as SQSTM1[34] or CD147[35], were detected in the MS analyses. As for the proteins involved in ubiquitination by CRL4[CRBN], biotinylation of DDB1, CUL4A, CUL4B, COP9 signalosome (CSN) components and UBC12 was detected, whereas that of RBX1, UBE2D3, and UBE2G1[36] was not detected (Supplementary Fig. 7a). In the case of proteins interacting with CRBN, biotinylation of SQSTM1[34], CD147[35], TP53[37], and HSP90AA[38] was detected in the MS analyses, while that of MEIS[39], MCT-1[35], CD98[38], and GS[40,41] was not detected (Supplementary Fig. 7b). In addition, biotinylation of some interacting proteins was increased or decreased, but not as greatly as that of the neo-substrates (Supplementary Fig. 7b). Although these results indicate that AirID-CRBN can biotinylate IMiDs-independent interactors, more suitable cell-lines or culture conditions may be required to analyse the CRBN interactors.

**AirID-CRBN with IMiDs presents neo-substrate candidates in cells**. Next, we attempted to identify neo-substrates from IMiDs-induced biotinylated proteins. CRBN interacts with proteins containing C2H2 zinc finger domain (C2H2-ZNF)[42], and the neo-substrate is subsequently degraded by the 26S proteasome. Therefore, we focused on biotinylated proteins containing ZNFs in the presence of IMiDs. In HEK293T cells, AirID-CRBN with IMiDs biotinylated several proteins containing C2H2-ZNF, such as ZNF629, ZNF644, and ZNF687 (Fig. 2a, b). Biotinylated peptides of DIDO1 (PHD-ZNF) and GATAD2A (GATA-ZBF) containing other types of ZNFs were detected (Fig. 2a, b). In addition, IMiDs-dependent biotinylation of ZNF687 was confirmed by STA-PDA (Fig. 2c), and endogenous ZNF687 was slightly degraded by IMiDs treatment (Fig. 2d). In HuH7 cells, seven proteins containing ZNF, such as ZNF629 and ZNF644, were detected by LC-MS/MS analysis of biotinylated peptides

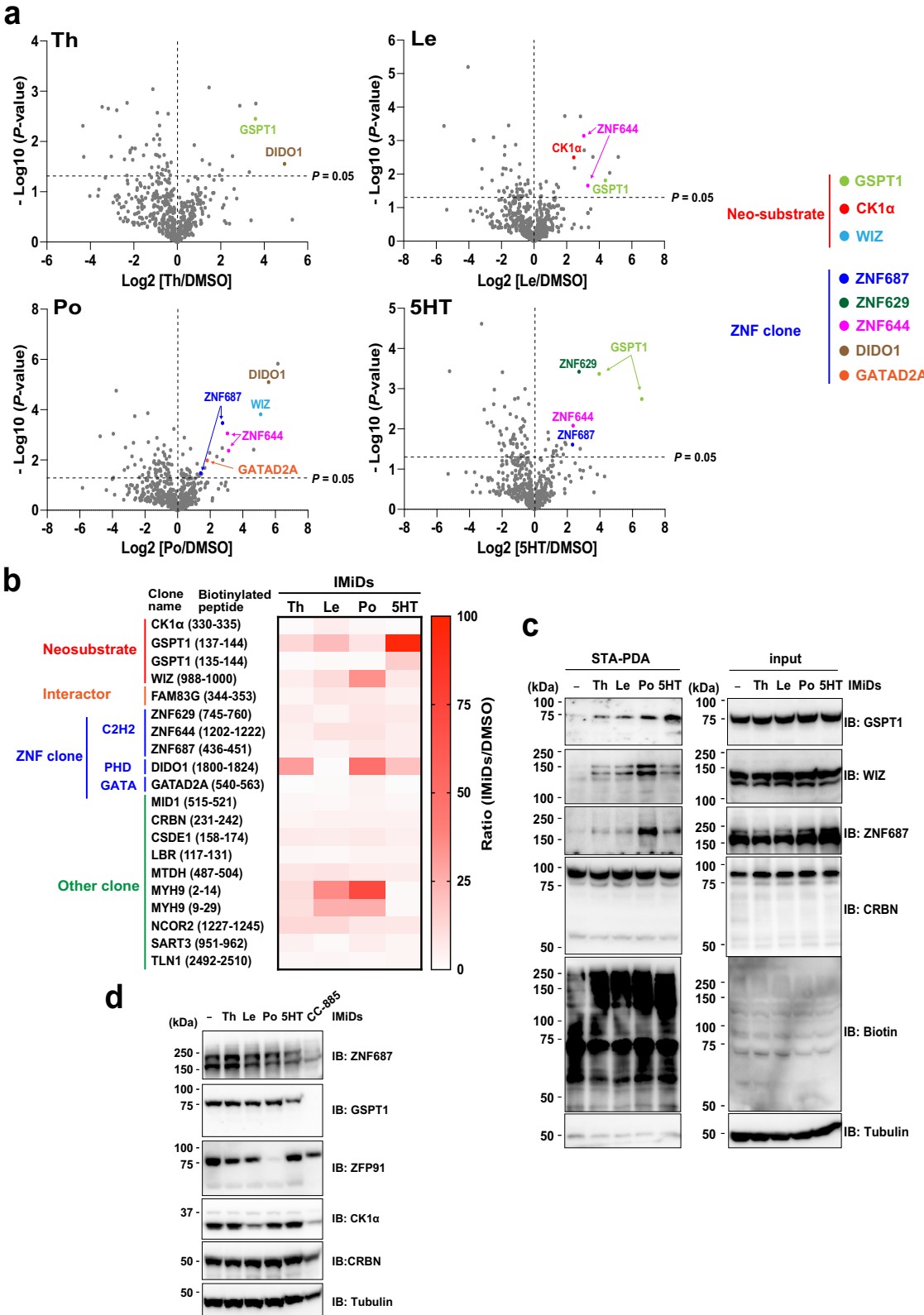

(Fig. 3c). In IMR32 cells, most of the biotinylated proteins were detected among all cell lines used in this study, and eleven proteins containing ZNF were detected, including ZNF536, ZNF687 and ZMYM2 (Fig. 3d). We then confirmed whether these candidates were biotinylated in the presence of IMiDs in HuH7 and IMR32 cells. Similar to the results of MS analysis, STA-PDA

showed that ZNF687 and ZMYM2 were biotinylated by IMiDs treatment (Fig. 3e, f).

BioID is thought to also biotinylate proteins of indirect interaction in multi-protein complexes; therefore, we examined whether these proteins directly interact with CRBN in the presence of IMiDs. In previous studies, we reported an

**Fig. 2 AirID-CRBN with thalidomide and IMiDs biotinylates neo-substrates in cells in a highly selective manner. a** LC-MS/MS analysis of biotinylated peptides using AirID-CRBN. HEK293T cells stably expressing AGIA-AirID-CRBN-WT were treated with DMSO, 20 μM thalidomide (Th), 10 μM lenalidomide (Le), 10 μM pomalidomide (Po) or 20 μM 5-hydroxythalidomide (5HT) in the presence of 10 μM biotin and 5 μM MG132 for 8 h (biological replicate; $n = 3$). **b** Heat map of biotinylated peptides (ThD/DMSO ratio >3 and $P$-value <0.05) in HEK293T cells detected by LC-MS/MS analysis. **a**, **b** Significant changes in the volcano plots and heat map were calculated by Student's two-sided $t$-test and the false discovery rate (FDR)-adjusted $P$-values calculated using Benjamini–Hochberg method are shown in the Supplementary Data 6. **c** IMiD-dependent biotinylation assay of neo-substrate candidates detected by LC-MS/MS analysis in HEK293T cells. HEK293T cells stably expressing AGIA-AirID-CRBN-WT were treated with DMSO, 20 μM thalidomide (Th), 10 μM lenalidomide (Le), 10 μM pomalidomide (Po) or 20 μM 5-hydroxythalidomide (5HT) in the presence of 10 μM biotin and 5 μM MG132 for 8 h. Then, the biotinylated proteins were pulled down using streptavidin beads and analysed by immunoblotting. The experiment was repeated three times independently with similar results. **d** Immunoblot analysis of endogenous protein levels of neo-substrates in HEK293T cells treated with DMSO, 10 μM thalidomide (Th), 10 μM lenalidomide (Le), 10 μM pomalidomide (Po), 10 μM 5-hydroxythalidomide (5HT) or 1 μM CC-885 for 24 h. The experiment was repeated twice independently with similar results. Source data are provided as a Source data file.

AlphaScreen-based biochemical assay using recombinant proteins synthesised by a wheat cell-free system[43] (Fig. 4a). We then performed an AlphaScreen-based biochemical assay using 20 μM IMiDs, because IMiDs over 10 μM were sufficient for detection of the interaction between CRBN and neo-substrates in our previous studies[11,30]. In the AlphaScreen assay, luminescence signals of ZNF536, ZNF687, and ZMYM2 increased upon IMiDs supplementation, but those of the other candidates did not (Fig. 4b). These results indicate that ZNF536, ZNF687, and ZMYM2 directly interacted with CRBN in the presence of IMiDs.

To investigate IMiDs-induced protein degradation of ZNF536, ZNF687, and ZMYM2, we constructed an expression vector of these proteins in cultured mammalian cells. Immunoblot analyses showed that pomalidomide induced protein degradation of Myc-ZNF687 and ZMYM2-AGIA, and 5HT induced protein degradation of Myc-ZNF687 (Supplementary Fig. 8a). By contrast, AGIA-ZNF536 was scarcely degraded by IMiDs treatment. Furthermore, ZNF687 and ZMYM2 induced protein degradation in a dose-dependent manner with pomalidomide and 5HT or pomalidomide alone, respectively (Fig. 4c). These findings suggest that ZNF687 and ZMYM2 are IMiDs-dependent neo-substrates of CRBN. Immunoblot analysis showed that pomalidomide induced protein degradation of endogenous ZMYM2 without downregulation of ZMYM2 mRNA expression, but not protein degradation of endogenous ZNF687 (Supplementary Fig. 8b, c and Fig. 4d, e). Furthermore, pomalidomide-dependent protein degradation of ZMYM2 was inhibited in the presence of MG132 and MLN4924, which are proteasome inhibitors and cullin E3 ligase inhibitors, respectively (Fig. 4e), and quantitative immunoblot analysis using fluorescent probes also showed that pomalidomide induced protein degradation of ZMYM2 (Fig. 4g). These results indicate that ZMYM2 is a neo-substrate of CRBN, and that pomalidomide selectively induces ZMYM2 protein degradation.

**Validation of BioID, TurboID and AirID enzymes for drug-inducible biotinylation.** Because it has been shown that the combination of AirID-CRBN and tamavidin 2-REV can identify neo-substrates (Figs. 1–4), we next compared AirID with BioID (BirA*) and TurboID[42], both of which are conventional proximity-dependent biotinylation enzymes. First, we generated HEK293T and IMR32 cells stably expressing BioID-CRBN and investigated whether BioID-CRBN can IMiDs-dependently biotinylate neo-substrates. In previous reports, the BioID enzyme was reported to require a long labelling time and high concentrations of biotin for biotinylation of interacting proteins[44,45]. Consistent with these reports, six hours of incubation with biotin and pomalidomide was insufficient for the biotinylation of neo-substrates (Fig. 5a, b). Because some drugs may show cytotoxicity to some cell-lines, such as IMiDs-treated multiple-myeloma cells,

more active enzymes would be ideal for drug-inducible biotinylation. Second, we generated HEK293T and IMR32 cells stably expressing TurboID-CRBN, and IMiDs-dependent biotinylation of neo-substrates was compared between AirID-CRBN and TurboID-CRBN. In immunoblot analyses of STA-PD, AirID-CRBN biotinylated ZMYM2, ZNF687, and PLZF after supplementation with pomalidomide and biotin for both 2 and 6 h (Fig. 5c, d). By contrast, TurboID-CRBN IMiDs-dependently biotinylated ZNF687 and PLZF, but not ZMYM2, after supplementation with pomalidomide and biotin for two hours (Fig. 5c, d). When pomalidomide and biotin were supplemented for six hours, pomalidomide-independent biotinylation was strongly observed in TurboID-CRBN-expressing cells (Fig. 5c, d).

To globally compare IMiDs-dependent biotinylation of neo-substrates between AirID-CRBN and TurboID-CRBN, we performed LC-MS/MS analyses following tamavidin 2-REV enrichment using IMR32 cells expressing AirID-CRBN or TurboID-CRBN under brief labelling conditions. Although almost half of the identified biotinylated proteins were the same, the other half were different (Fig. 5e, Supplementary Data 9, 10). In the absence of IMiDs (DMSO), TurboID-CRBN biotinylated more proteins (759 proteins) than AirID-CRBN (642 proteins). However, in the presence of pomalidomide, AirID-CRBN biotinylated more proteins (626 proteins) than TurboID-CRBN (584 proteins). The difference in biotinylated proteins might be due to their enzymatic nature. Both AirID-CRBN and TurboID-CRBN biotinylated some neo-substrates, such as SALL4 and WIZ, in a pomalidomide dependent manner (Fig. 5f, Supplementary Data 9, 10). However, pomalidomide-dependent biotinylation of ZMYM2 was observed only in AirID-CRBN-expressing cells, and that of ZFP91 and ZNF827 was observed only in TurboID-CRBN-expressing cells (Fig. 5f, Supplementary Data 9, 10). Taken together, these results suggest that, although they have different properties, both AirID and TurboID are useful enzymes for the detection of IMiDs-inducible biotinylation of neo-substrates under brief labelling conditions.

**ZMYM2 interacts with CRBN via MYM-ZNF for selective degradation by pomalidomide.** As described in Fig.4, CRBN mainly interacts with neo-substrates in the presence of IMiDs via C2H2-ZNF, leading to protein degradation of the neo-substrates[6,7,9–11,42]. Importantly, ZMYM2 does not have C2H2-ZNF and has an MYM-type zinc finger domain (MYM-ZNF) (Fig. 6a). Since ZMYM family proteins have an MYM-ZNF (Fig. 6a), we investigated whether other ZMYM proteins interact with CRBN in the presence of IMiDs via the AlphaScreen assay. ZMYM3 and 4, which are the most similar proteins to ZMYM2 in the family, did not interact with CRBN (Fig. 6b). To examine the interaction region in ZMYM2, we synthesised three truncated proteins (Supplementary Fig. 9a). The AlphaScreen assay showed

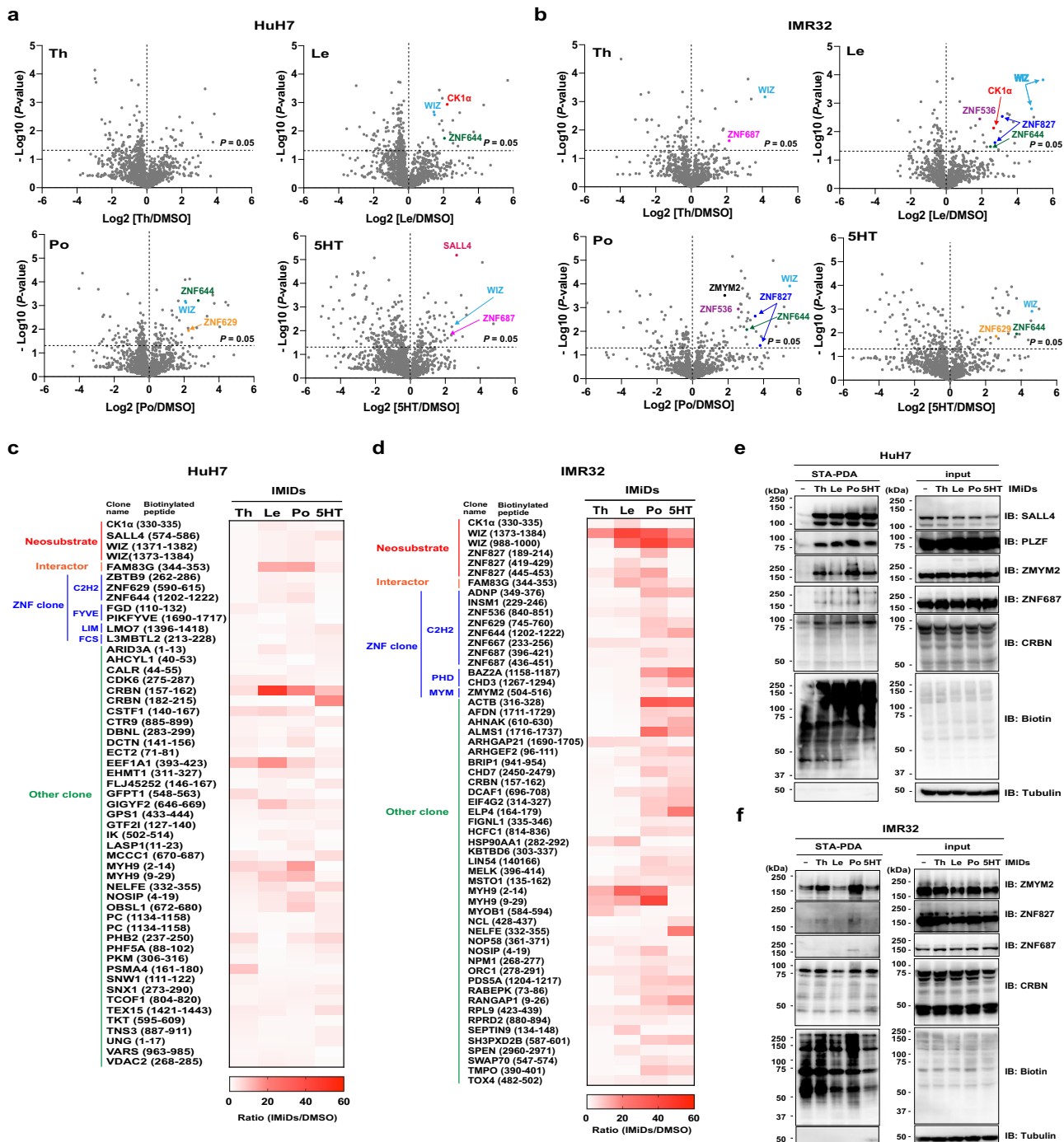

**Fig. 3 Biotin-dependent LS-MS/MS analyses using HuH7 and IMR32 cells expressing AirID-CRBN. a**, **b** LC-MS/MS analysis of biotinylated peptide using AirID-CRBN for (**a**) HuH7 or (**b**) IMR32 cells. HuH7 or IMR32 cells stably expressing AGIA-AirID-CRBN-WT were treated with DMSO, 20 μM thalidomide (Th), 10 μM lenalidomide (Le), 10 μM pomalidomide (Po) or 20 μM 5-hydroxythalidomide (5HT) in the presence of 10 μM biotin and 5 μM MG132 for 8 h (biological replicates; n = 3). **c**, **d** Heat map of biotinylated peptides (IMiDs/DMSO ratio >3 and P-value <0.05) in **c** HuH7, or **d** IMR32 cells detected by LC-MS/MS analysis. **a**–**d** Significant changes in the volcano plots and heat maps were calculated by Student's two-sided t-test and the false discovery rate (FDR)-adjusted P-values calculated using Benjamini–Hochberg method are shown in the Supplementary Data 7, 8. **e**, **f** IMiDs-dependent biotinylation assay of neo-substrate candidates detected by LC-MS/MS analyses in **e** HuH7 or **f** IMR32 cells. HuH7 or IMR32 cells stably expressing AGIA-AirID-CRBN-WT were treated with DMSO, 20 μM thalidomide (Th), 10 μM lenalidomide (Le), 10 μM pomalidomide (Po) or 20 μM 5-hydroxythalidomide (5HT) in the presence of 10 μM biotin and 5 μM MG132 for 8 h. Then, the biotinylated proteins were pulled down using streptavidin beads and analysed by immunoblotting. The experiments were repeated twice independently with similar results. Source data are provided as a Source data file.

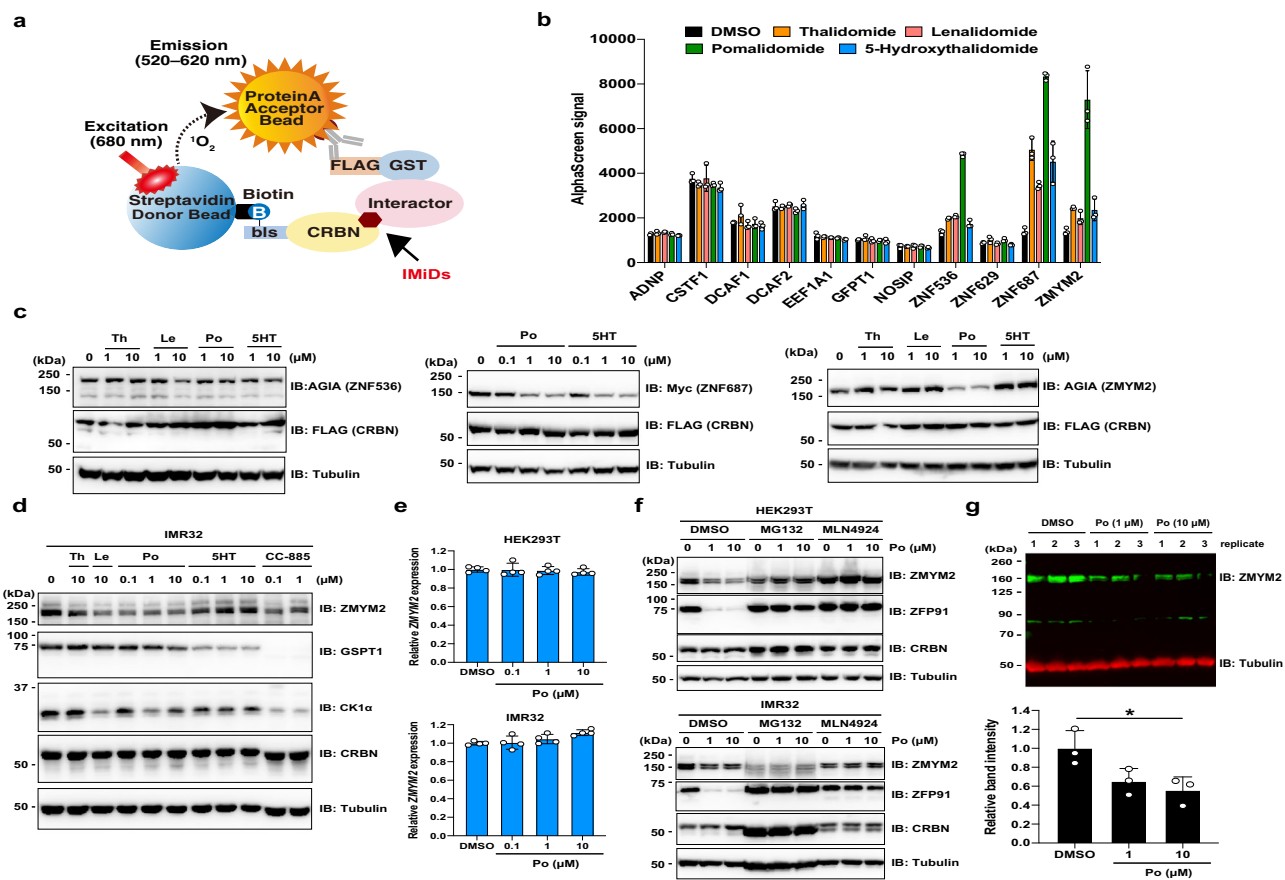

**Fig. 4 AirID-CRBN with thalidomide and IMiDs presents neo-substrate candidates in cells. a** Schematic diagram of AlphaScreen-based biochemical assay for detecting IMiDs-dependent interaction between CRBN and neo-substrates. **b** In vitro interaction assay between CRBN and neo-substrate candidates. The interaction between biotin-labelled bls-CRBN and FLAG-GST-neo-substrate candidates in the presence of DMSO or 20 μM thalidomide (Th), lenalidomide (Le), pomalidomide (Po) or 5-hydroxythalidomide (5HT) was analysed using the AlphaScreen-based biochemical assay. All AlphaScreen signals mean raw luminescent signal in the AlphaScreen-based biochemical assay. Error bars denote the standard deviation (independent experiments; n = 3). **c** Immunoblot analysis of IMiD-dependent protein degradation of ZNF536, ZNF687 and ZMYM2. HEK293T cells expressing AGIA-ZNF536, Myc-ZNF687 or ZMYM2-AGIA and FLAG-CRBN were treated with DMSO, thalidomide (Th), lenalidomide (Le), pomalidomide (Po) or 5-hydroxythalidomide (5HT) for 16 h. The experiment was repeated three times independently with similar results. **d** Immunoblot analysis of endogenous ZMYM2 protein levels in IMR32 cells treated with DMSO, thalidomide (Th), lenalidomide (Le), pomalidomide (Po), 5-hydroxythalidomide (5HT) or CC-885 for 24 h. The experiment was repeated three times independently with similar results. **e** IMR32 and HEK293T cells were treated with DMSO or pomalidomide (Po) for 24 h and ZMYM2 mRNA expression levels were measured by quantitative RT-PCR. Relative mRNA expression was determined using the expression level with DMSO treatment. Error bars denote the standard deviation (biological replicates; n = 4). **f** Immunoblot analysis of ZMYM2 protein levels in IMR32 and HEK293T cells treated with DMSO or pomalidomide (Po) in the presence of DMSO, 5 μM MG132 or 1 μM MLN4924 for 24 h. **g** Immunoblot analysis of ZMYM2 protein levels in IMR32 cells treated with DMSO or pomalidomide (Po) for 48 h ZMYM2 band intensity was normalised to tubulin band intensity. All relative band intensities are expressed as fluorescent signals relative to that of DMSO. Error bars denote the standard deviation (biological replicates; n = 3) and P-values were calculated by one-way ANOVA with Tukey's post-hoc test (*P = 0.031). Source data are provided as a Source data file.

that ZMYM2 interacts with CRBN via N-terminal MYM-ZNF (Supplementary Fig. 9b). Furthermore, we narrowed down a key MYM-ZNF for interaction with CRBN by synthesising truncated MYM-ZNF proteins (Supplementary Fig. 9c). The AlphaScreen assay indicated that ZMYM2 interacts with CRBN in the presence of pomalidomide via the third MYM-ZNF (Supplementary Fig. 9d). Next, we constructed mutant ZMYM2 proteins that substituted Cys470 to Ala to carry out biochemical and cellular degradation assays. These results revealed that the third MYM-ZNF in ZMYM2 is required for the interaction with CRBN (Fig. 6c, d).

**ZMYM2-FGFR1 fusion protein is degraded by pomalidomide.**
ZMYM2 was first characterised as a protein responsible for the induction of 8p11 myeloproliferative syndrome (EMS)/stem cell

leukaemia-lymphoma (SCLL) by chromosomal translocation to FGFR1[46,47], resulting in rapid transformation to acute myeloid leukaemia (AML) and T-lymphoblastic lymphoma[46,47]. ZMYM2-FGFR1 comprises an N-terminal MYM-ZNF in ZMYM2 and a C-terminal kinase domain in FGFR1[46,47] (Fig. 6e). Because ZMYM2 interacts with CRBN via the third MYM-ZNF, we investigated whether ZMYM2-FGFR1 also interacts with CRBN in the presence of pomalidomide. As expected, the AlphaScreen-based biochemical assay showed that pomalidomide-dependent interaction between ZMYM2-FGFR1 and CRBN, but not FGFR1, was detected (Fig. 6f). Immunoblot analysis showed that transiently expressed AGIA-ZMYM2-FGFR1 was degraded by pomalidomide treatment in HEK293T cells, but FGFR1 was not (Supplementary Fig. 10a). Furthermore, it was confirmed that pomalidomide selectively induced protein degradation of AGIA-ZMYM2-FGFR1 in HEK293T cells (Supplementary Fig. 10b). In

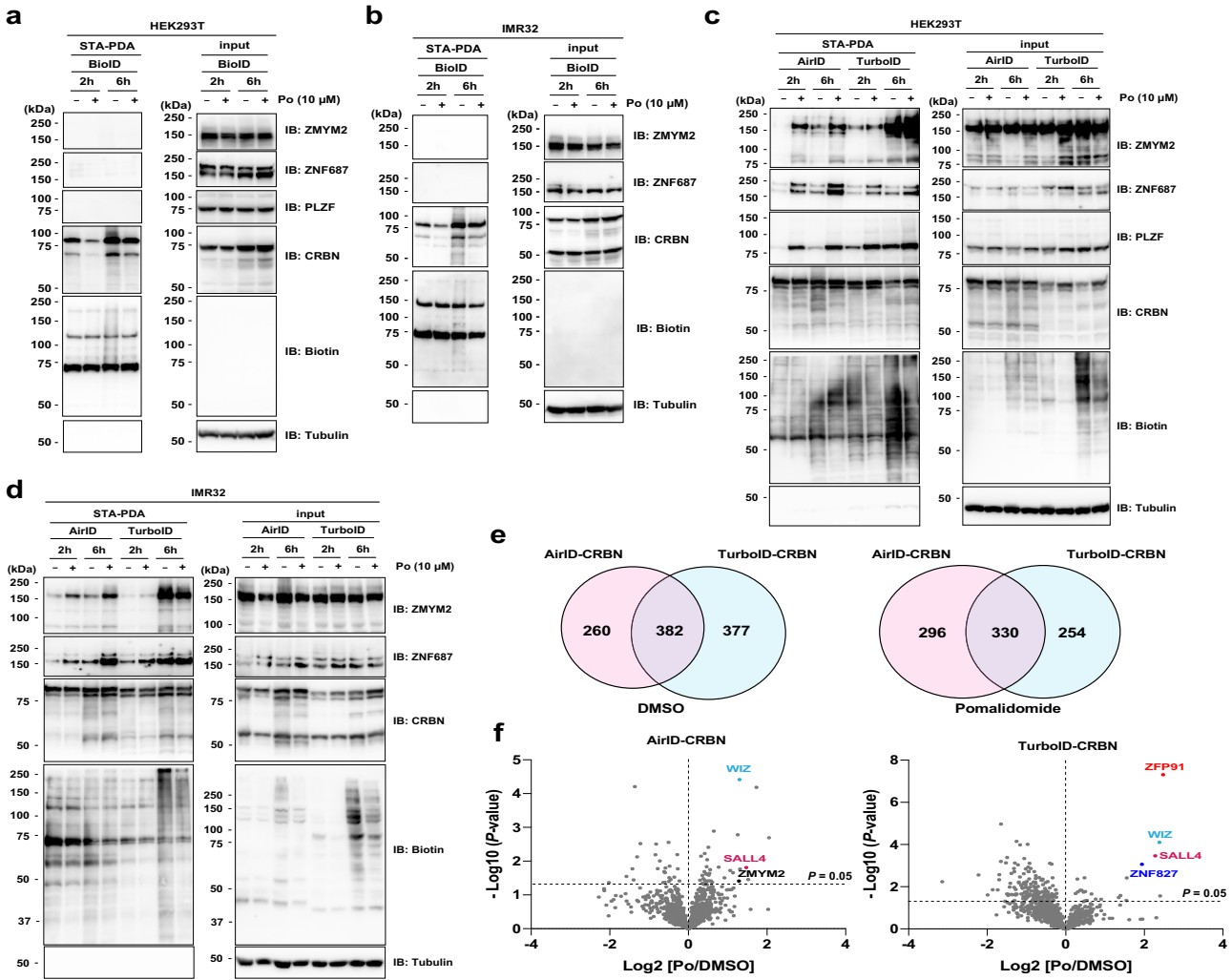

**Fig. 5 Comparison of biotinylation enzymes in biotinylation of neo-substrate. a, b** IMiDs-dependent biotinylation assay of neo-substrates by BioID-CRBN in **a** HEK293T or **b** IMR32 cells. HEK293T or IMR32 cells stably expressing AGIA-BioID-CRBN were treated with DMSO or 10 μM pomalidomide (Po) in the presence of 10 μM biotin and 5 μM MG132 for 2 h or 6 h. The experiments were repeated twice independently with similar results. **c, d** IMiDs-dependent biotinylation assay of neo-substrates by AirID-CRBN or TurboID-CRBN in **c** HEK293T or **d** IMR32 cells. HEK293T or IMR32 cells stably expressing AGIA-AirID-CRBN or AGIA-TurboID-CRBN were treated with DMSO or 10 μM pomalidomide (Po) in the presence of 10 μM biotin and 5 μM MG132 for 2 h or 6 h. The experiments were repeated twice independently with similar results. **e** Comparison between AirID-CRBN and TurboID-CRBN of biotinylated proteins detected by LC-MS/MS analysis using IMR32 cells. Overlapping biotinylated proteins were compared between AirID-CRBN and TurboID-CRBN using a Venn diagram. **f** LC-MS/MS analysis of biotinylated peptides using AirID-CRBN or TurboID-CRBN in IMR32 cells. IMR32 cells stably expressing AGIA-AirID-CRBN or AGIA-TurboID-CRBN were treated with DMSO or 10 μM pomalidomide (Po) in the presence of 10 μM biotin and 5 μM MG132 for 2 h (biological replicates; n = 3). Significant changes in the volcano plots were calculated by Student's two-sided t-test and the false discovery rate (FDR)-adjusted P-values calculated using Benjamini–Hochberg method are shown in the Supplementary Data 9, 10. **a–d** Biotinylated proteins were pulled down using streptavidin beads and analysed by immunoblotting. Source data are provided as a Source data file.

previous reports, ZMYM2-FGFR1 continuously activates the kinase domain in the C-terminal and FGFR signalling, resulting in STAT1/3 phosphorylation and the induction of diverse genes involved in cell survival and proliferation by ERK1/2 in cancer cells[48–50]. Therefore, we investigated whether pomalidomide-induced protein degradation of ZMYM2-FGFR1 affected FGFR signalling activation. First, we constructed HEK293T cells stably expressing ZMYM2-FGFR1 by lentivirus infection. Consistent with the previous reports, ZMYM2-FGFR1 induced the phosphorylation of STAT1/3 and ERK1/2 (Supplementary Fig. 10c). In stable cells, pomalidomide treatment reduced the phosphorylation levels of STAT1/3 and ERK with ZMYM2-FGFR1 degradation (Fig. 6g). Then, we constructed stable cells expressing ZMYM2-FGFR1-C470A, which is unbound by CRBN (Supplementary Fig. 10d), by infection with lentivirus.

Immunoblot analysis showed that pomalidomide treatment reduced phosphorylated STAT1/3 and ERK in cells stably expressing ZMYM2-FGFR1-WT (WT), but not in cells expressing ZMYM2-FGFR1-C470A (C470A; Fig. 6h). These results strongly suggest that pomalidomide attenuates the STAT1/3 and ERK signalling cascades by degradation of ZMYM2-FGFR1. Taken together, our results showed that pomalidomide could be a candidate drug for treating cancers driven by ZMYM2 and FGFR1 fusion, such as EMS.

**AirID-fusion approach can be applied to molecular glues and PROTACs.** Indisulam is an aryl sulphonamide drug (Fig. 7a) that exerts anticancer effects on several human cell types[51,52]. Indisulam is a second class of molecular glue degraders in mammalian

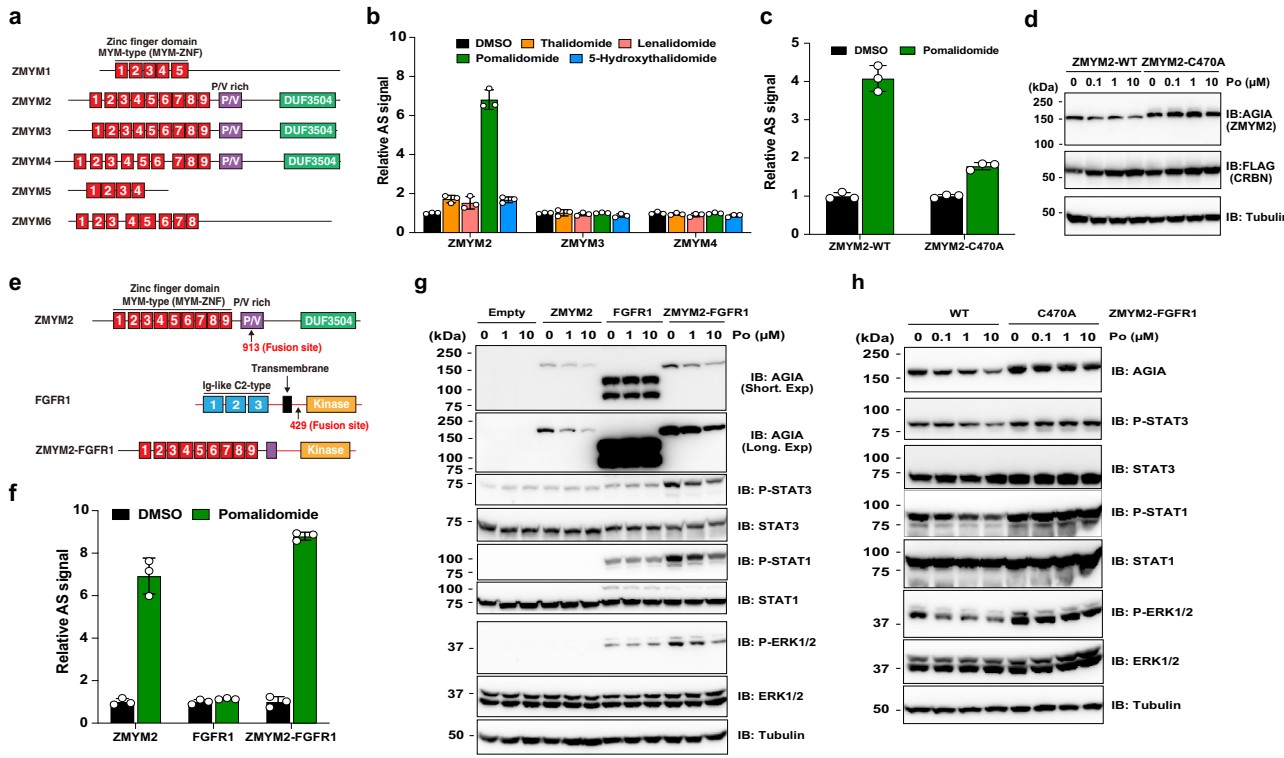

**Fig. 6 ZMYM2-FGFR1 fusion protein is degraded by pomalidomide. a** Schematic diagram of ZMYM family proteins. **b** In vitro interaction assay between CRBN and ZMYM2, ZMYM3 or ZMYM4. The interaction between bls-CRBN and FLAG-GST-ZMYM2, -ZMYM3 or -ZMYM4 in the presence of DMSO or 20 μM thalidomide (Th), lenalidomide (Le), pomalidomide (Po) or 5-hydroxythalidomide (5HT) was analysed using the AlphaScreen-based biochemical assay. All relative AlphaScreen (AS) signals were expressed as a luminescence signal relative to that of DMSO. Error bars denote the standard deviation (independent experiments; $n = 3$). **c** In vitro interaction assay between CRBN and ZMYM2-WT or ZMYM2-C470A. The interaction between bls-CRBN and FLAG-GST-ZMYM2-WT or -C470A in the presence of DMSO or 20 μM pomalidomide (Po) was analysed using the AlphaScreen-based biochemical assay. All relative AlphaScreen (AS) signals were expressed as a luminescence signal relative to that of DMSO. Error bars denote the standard deviation (independent experiments; $n = 3$). **d** Immunoblot analysis of ZMYM2-AGIA protein levels in HEK293T cells expressing ZMYM2-WT- or ZMYM2-C470A-AGIA and FLAG-CRBN treated with DMSO or pomalidomide (Po) for 16 h. The experiments were repeated three times independently with similar results. **e** Schematic diagram of ZMYM2, FGFR1, ZMYM2-FGFR1 protein. **f** In vitro interaction assay between CRBN and ZMYM2, FGFR1 or ZMYM2-FGFR1. The interaction between bls-CRBN and FLAG-GST-ZMYM2, -FGFR1 or -ZMYM2-FGFR1 in the presence of DMSO or 20 μM pomalidomide (Po) was analysed using the AlphaScreen-based biochemical assay. All relative AlphaScreen (AS) signals were expressed as a luminescence signal relative to that of DMSO. Error bars denote the standard deviation (independent experiments; $n = 3$). **g** Immunoblot analysis of protein levels in HEK293T cells stably expressing empty, ZMYM2-WT- FGFR1- or ZMYM2-C470A-AGIA treated with DMSO or pomalidomide (Po) every 24 h for 48 h. The experiments were repeated three times independently with similar results. **h** Immunoblot analysis of protein levels in HEK293T cells stably expressing empty, ZMYM2-FGFR1-WT-AGIA (WT) or ZMYM2-C470A-AGIA (C470A) treated with DMSO or pomalidomide (Po) every 24 h for 48 h. The experiments were repeated three times independently with similar results. Source data are provided as a Source data file.

cells, and exerts anticancer effects by degrading RNA-binding protein 39 (RBM39) by interacting with DCAF15 (DDB1- and CUL4-associated factor 15)[14,15]. It was reported that DCAF15-BioID biotinylated RBM39 in K562 cells treated with indisulam for 72 h[45]. As shown in Fig. 5, AirID-CRBN biotinylated the neo-substrate with a shorter labelling time (~2 h); therefore, we examined whether the AirID method could be applied to indisulam (Fig. 7b). First, we generated HCT116 cells stably expressing AGIA-AirID-DCAF15-WT or -D475N (unbound mutant with indislam[53]) by lentivirus infection. STA-PDA and immunoblot analysis showed that AGIA-AirID-DCAF15-WT biotinylated endogenous RBM39 in an indisulam-dependent manner, but AGIA-AirID-DCAF15-D264N did not (Fig. 7c). Furthermore, LC-MS/MS analysis of biotinylated peptides revealed that RBM39 was significantly biotinylated in the presence of indisulam in HCT116 cells (Fig. 7d, Supplementary Data 11). These results indicate that the AirID-fusion E3 ligase can be applied to research involving molecular glues.

PROTACs are drugs synthesised from two functional compounds: an E3 ligase binder and a target binder[16,17]. Many IMiDs-based PROTACs have been developed for therapy because of their high drug action, which induces thorough degradation of target protein[16,17]. However, IMiDs-based PROTACs induce degradation of not only target proteins, but also neo-substrates of IMiDs, presenting a large issue with their application[9]. Consistent with previous reports, the thalidomide- and pomalidomide-based PROTACs for BRD4 (dBET1 and ARV-825, respectively; Fig. 7e) induced protein degradation of neo-substrates such as IKZF1, IKZF3, and SALL4 (Supplementary Fig. 11a). This degradation is thought to occur due to high drug action or adverse effects. For example, protein degradation of IKZF1 and IKZF3, in addition to BRD4, inhibits MM cell growth (Supplementary Fig. 11a, b). On the other hand, there is concern that protein degradation of SALL4 and PLZF causes teratogenic effects on foetuses in pregnant women (Supplementary Fig. 11a)[9]. Therefore, we attempted to use a combination of AirID-CRBN and biotin-dependent LC-MS/MS analysis for PROTACs validation. STA-PDA showed that AirID-CRBN could biotinylate BRD4, IKZF1, and IKZF3 by dBET1 or ARV-825 treatment in MM1.S cells (Fig. 7f). High doses of PROTACs lead to a "hook effect" and

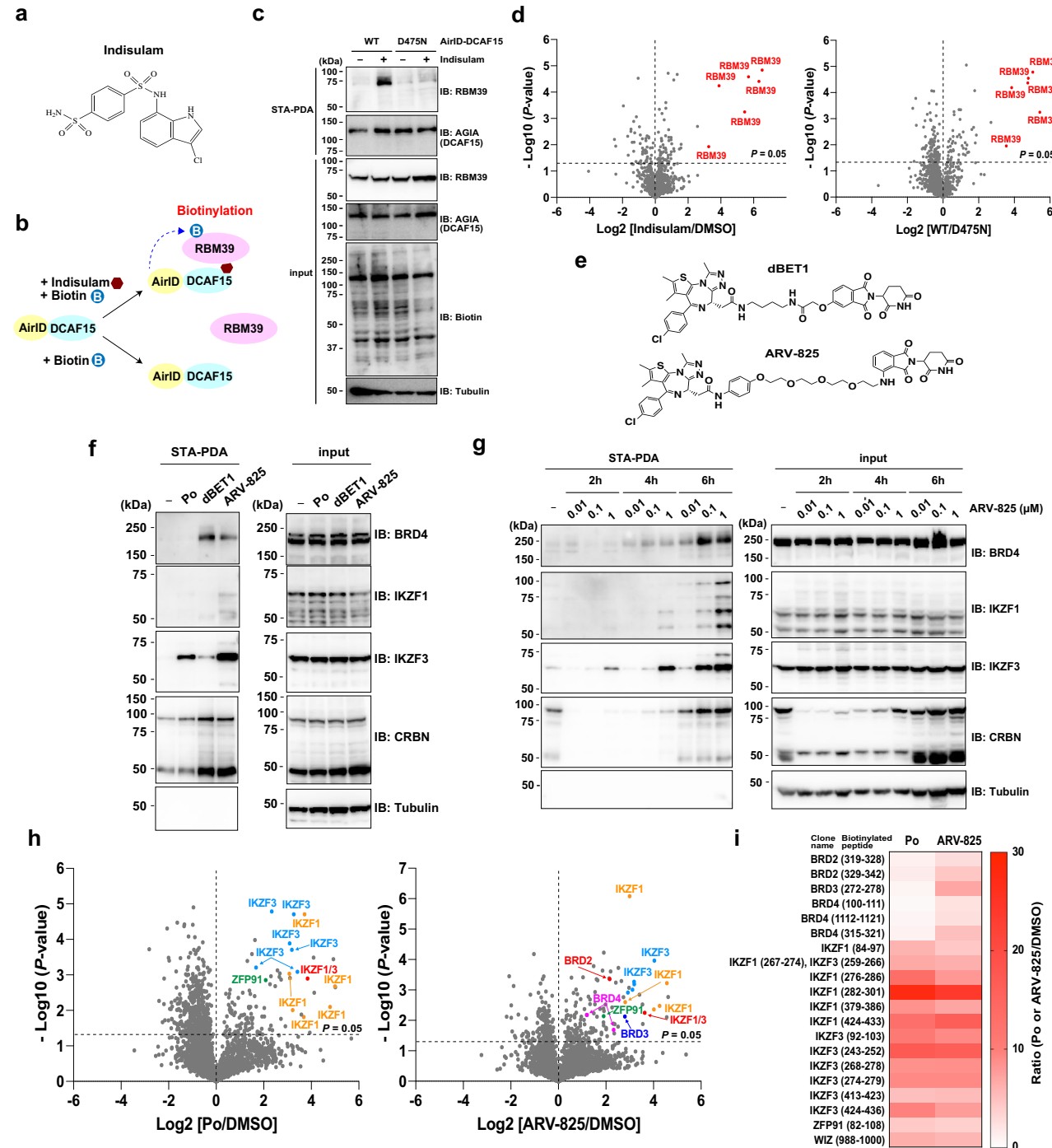

cannot induce target protein degradation; thus, the dose of PROTACs is very important for yielding efficient effects[54]. AlphaScreen-based biochemical and protein degradation assays showed that high doses of dBET1 or ARV-825 cannot form a CRBN-PROTACs-BRD2/3/4 complex (Supplementary Fig. 11c–e). Therefore, we first examined the appropriate treatment time and dose of PROTACs for efficient biotinylation of BRD4 by AirID-CRBN. STA-PDA revealed that a high PROTACs dose and a long treatment time induced biotinylation of IKZF1 and IKZF3 rather than BRD4 (Fig. 7g). Furthermore, it was confirmed that a biotin dose of 5 μM was optimal for biotinylation of BRD4 by STA-PDA (Supplementary Fig. 11f). Based on these results, we determined that 6 h of treatment at 0.1 μM ARV-825 and 5 μM biotin was optimal. We then performed biotin-dependent LC-MS/MS analysis using DMSO, pomalidomide, or ARV-825. Biotinylated peptides of BRD2, BRD3, and BRD4 were detected in ARV-825-treated lysates, but not in DMSO- or pomalidomide-treated lysates (Fig. 7h, i, Supplementary Data 12). By contrast, biotinylated peptides of IKZF1 and IKZF3 were detected in both the pomalidomide and ARV-825 lysates (Fig. 7h, i). These results suggest that the AirID system can be applied to research on PROTACs for analysing drug-inducible PPIs.

## Discussion

Due to the clinical success of the thalidomide derivatives—lenalidomide and pomalidomide—molecular glue degraders are attractive drugs for the treatment of many diseases[55,56]. IMiDs—such as thalidomide, lenalidomide, and pomalidomide—are the

**Fig. 7 AirID can be applied to other molecular glues and PROTACs. a** Chemical structure of indisulam. **b** Schematic diagram of indisulam-dependent biotinylation of RBM39 by AirID-DCAF15. **c** Indisulam-dependent biotinylation of RBM39 in HCT116 cells. HCT116 cells stably expressing AGIA-AirID-DCAF15-WT or -D475N were treated with DMSO or 5 μM indisulam and 50 μM biotin and 10 μM MG132 for 6 h. The experiments were repeated twice independently with similar results. **d** LC-MS/MS analysis of indisulam-dependent biotinylation in HCT116 cells. HCT116 cells stably expressing AGIA-AirID-DCAF15-WT or -D475N were treated with DMSO or 5 μM indisulam in the presence of 50 μM biotin and 10 μM MG132 for 6 h (biological replicates; n = 3). Significant changes in the volcano plots were calculated by Student's two-sided t-test and the false discovery rate (FDR)-adjusted P-values calculated using Benjamini–Hochberg method are shown in Supplementary Data 11. **e** Chemical structures of dBET1 and ARV-825. **f** Pomalidomide (Po)- or PROTACs-dependent biotinylation assay of neo-substrates and BRD4 in MM1.S cells. MM1.S cells stably expressing AGIA-AirID-CRBN-WT were treated with DMSO, 1 μM Po, 10 μM dBET1 or 0.1 μM ARV-825 in the presence of 10 μM biotin and 5 μM MG132 for 8 h. The experiments were repeated three times independently with similar results. **g** Time course and dose-dependent analyses of PROTACs-induced biotinylation of neo-substrates and BRD4 in MM1.S cells. MM1.S cells stably expressing AGIA-AirID-CRBN-WT were treated with DMSO or ARV-825 (0.01, 0.1, or 1 μM) in the presence of 10 μM biotin and 5 μM MG132 for 2, 4 or 6 h. The experiments were repeated twice independently with similar results. **h** LC-MS/MS analysis of PROTACs-dependent biotinylation in MM1.S cells. MM1.S cells stably expressing AGIA-AirID-CRBN-WT were treated with DMSO, 1 μM pomalidomide (Po) or 0.1 μM ARV-825 in the presence of 5 μM biotin and 5 μM MG132 for 6 h (biological replicates; n = 3). Significant changes in the volcano plots were calculated by Student's two-sided t-test and the false discovery rate (FDR)-adjusted P-values calculated using Benjamini–Hochberg method are shown in Supplementary Data 12. **i** Heat map of biotinylated peptides of known neo-substrates and ARV-825-targeted protein in MM1.S cells detected by LC-MS/MS analysis. **c, f, g** Biotinylated proteins were pulled down using streptavidin beads and analysed by immunoblotting. Source data are provided as a Source data file.

most thoroughly characterised molecular glue degraders, and many neo-substrates involved in anti-MM or -teratogenic effects have been identified[6,7,9–11]. Cell-based screening systems have previously been used for the identification of neo-substrates[6–8,10,31]. One such system is mass spectrometry using stable isotope labelling with amino acids in cell culture (SILAC), which is most broadly used for identifying neo-substrates[6,8]. In addition to cell-based systems, in vitro[11] or in silico[42] screening systems have been applied, although the actions of several drugs and their adverse effects, including sedative effects, remain unclear. Furthermore, thalidomide derivatives such as CC-122[57], CC-220[58], and CC-90009[59] have been actively developed as drugs for many diseases. Therefore, drug-inducible PPI analysis of cells is an important avenue of research for both the use and development of IMiDs. Herein, we developed a cell-based method for molecular glues- and PROTACs-inducible interactions using a combination of AirID and LC-MS/MS analysis of biotinylated peptides. Importantly, it was shown that AirID has higher drug-dependent biotinylation activity than conventional biotinylation enzymes such as BioID and TurboID (Fig. 5). In addition, the tamavidin 2-REV method[22] enabled efficient enrichment of biotinylated peptides without undesirable, non-specific contamination (Fig. 1d). These analyses showed that this method is suitable for analysing multiple drug-inducible interactions with high selectivity and specificity. Therefore, we believe our approach will provide a methodology for elucidating the molecular mechanisms of not only molecular glues and PROTACs, but also small molecules such as phytohormones.

In this study, we identified ZMYM2 as a selective neo-substrate of pomalidomide (Fig. 4 and Supplementary Fig. 8). Many neo-substrates of IMiDs are transcription factors containing C2H2 zinc finger domain (C2H2-ZNF) and interact with CRBN through C2H2-ZNF[31]. However, C2H2-ZNF was not identified in ZMYM2. Our biochemical and cell-based analyses revealed that ZMYM2 interacts with CRBN through MYM-type ZNF (MYM-ZNF) (Fig. 6a–d and Supplementary Fig. 9). Therefore, it is possible that there are unknown neo-substrates containing other types of ZNFs. Although many analyses are required to identify such neo-substrates, our approach can mitigate these challenges. ZMYM2 reportedly forms the ZMYM2-FGFR1 fusion protein and continuously activates FGFR1 signalling, resulting in AML[60]. ZMYM2-FGFR1–driven AML is a rare case but ZMYM2 is the most frequent as FGFR1 partner gene[60,61]. Our results showed that ZMYM2-FGFR1 interacts with CRBN in the presence of pomalidomide, which subsequently leads to the protein

degradation of ZMYM2-FGFR1 (Fig. 6f and Supplementary Fig. 10a, b). In more detailed analyses using HEK293T cells expressing ZMYM2-FGFR1, pomalidomide-dependent protein degradation of ZMYM2-FGFR1 reduced the levels of phosphorylated STAT1/3 and ERK1/2 (Fig. 6g, h). Very recently, Ebert et al. reported that avadomide (CC-122) induces protein degradation of ZMYM2 more strongly than does pomalidomide[62]. Furthermore, avadomide treatment reduced the number of viable bone marrow CD34+ cells isolated from ZMYM2-FGFR1-positive hematologic malignancy patients[62]. Therefore, their report and our results suggest that pomalidomide, not only avadomide, may also be effective drugs for ZMYM2-FGFR1–driven AML.

Our results showed that ZNF536 and ZNF687 directly interacted with CRBN but were not degraded (Fig. 4 and Supplementary Fig. 8). Previously, Thomä et al. identified 11 C2H2-ZNFs that were degraded in C2H2-ZNF library screening and concluded that zinc fingers have a low degree of sequence similarity and that only distinct amino acid combinations at the drug–CRBN interface resulted in protein degradation[42]. Furthermore, they revealed that docking analysis based on crystal structure information can predict C2H2-ZNFs that interact with CRBN-IMiDs with high probability[42]. Therefore, we confirmed the score of the zinc fingers in ZNF536 or ZNF687 based on the data from in silico analyses[42]. As a result, all zinc fingers in ZNF536 and ZNF687 showed low ranking in their in silico analyses. It has been reported that lenalidomide selectively induces protein degradation of CK1α, although CK1α interacts with CRBN in the presence of pomalidomide and thalidomide[32]. In fact, it was confirmed that other thalidomide derivatives induced protein degradation of C2H2-ZNFs, which interact with but are not degraded by CRBN–pomalidomide. Therefore, it is possible that ZNF536 and ZNF687 are degraded in the presence of other thalidomide derivatives. However, further analyses are required to understand whether ZNF536 and ZNF687 are degraded by such thalidomide derivatives or interact with CRBN-IMiDs without being degraded.

PROTACs are the alternative technology for inducing degradation of target proteins, and many IMiDs-based PROTACs have been developed[16,17]. Because PROTACs induces degradation of target proteins to below the detection limit, the drug action of PROTACs is quite drastic. However, IMiDs-based PROTACs reportedly induce protein degradation of the neo-substrate of IMiDs in addition to the target protein[9]. It is thought that this double degradation of the target protein and neo-substrate has both advantages and disadvantages. For example, it is possible

that SALL4 or PLZF degradation by IMiDs-based PROTACs causes adverse teratogenic effects. Furthermore, if chemical compounds interacting with target proteins bind to other proteins, the effect of protein degradation must be considered. Therefore, understanding the proteins that interact with CRBN by PROTACs treatment is very important for the development of PROTACs. Using IMiDs-based PROTACs for BRD family proteins, we showed that the method developed in this study can be applied to PROTACs (Fig. 7f–i).

As it is expected that this method can identify degradation-independent interactors, this system using AirID and biotin-dependent MS analysis is a unique approach for analysing target E3 ubiquitin ligase, molecular glues, and PROTACs. Protein degradation of neo-substrates in cells plays a pivotal role in the effects of molecular glues; however, it is possible that natural ligands of CRBN or non-degradative IMiDs-dependent interactors are also important for understanding the mechanism of action of molecular glues. Natural ligands are reportedly involved in IMiDs-dependent phenotypes in the previous studies[35,38,39]. Several components of CRL4, such as CUL4A/B and DDB1, and several CRBN interactors, such as SQSTM1 and HSP90, were detected by LC-MS/MS analysis in this study (Supplementary Fig. 7a, b). By contrast, biotinylated peptides of several interactors —such as MEIS2, CD98, and GL2—were not detected in these analyses. To detect biotinylated peptides of these interactors, other cell lines or culture conditions may be required. Furthermore, AirID-fused proteins can biotinylate indirect interactions, such as components of protein complexes. For example, biotinylated peptides of ZNF644, which is a partner protein of the neo-substrate WIZ[63], were detected in HEK293T, HuH7, and IMR32 cells (Figs. 2a, 3a, b). In addition, ZNF629 was identified by biotin-dependent LC-MS/MS analysis in HEK293T, HuH7, and IMR32 cells (Figs. 2a, 3a, b); however, the AlphaScreen-based biochemical assay did not show that ZNF629 is a direct IMiDs-dependent interactor with CRBN (Fig. 4b). Therefore, we believe that this method provides a powerful approach for understanding the overall drug-dependent interactome, including natural interactors of target proteins, degradative proteins, and non-degradative proteins.

In conclusion, the AirID fusion approach developed in this study is a robust method for understanding the drug-inducible interactomes.

## Methods

**Reagents**. Thalidomide (Tokyo Chemical Industry), pomalidomide (Sigma-Aldrich), lenalidomide (FUJIFILM Wako), 5-hydroxythalidomide (prepared according to a previously published method[64]), dBET1 (MedChemExpress), ARV-825 (MedChemExpress), indisulam (BLD Pharmatech), MG132 (Peptide Institute), and MLN4924 (Chemscene) were dissolved in DMSO (FUJIFILM Wako) at 2–100 mM and stored at −20 °C as stock solutions. All drugs were diluted 1000-fold for in vivo experiments or 200-fold for in vitro experiments.

**Plasmids**. The pDONR221, pcDNA3.1(+), or pCAGGS vectors were purchased from Invitrogen or RIKEN, respectively. The pEU vector for wheat cell-free protein synthesis was constructed in our laboratory, as previously described[43]. The pcDNA3.1(+)-FLAG-GW, pcDNA3.1(+)-Myc-MCS, pcDNA3.1(+)-AGIA-MCS, pCAGGS-MCS-AGIA, pEU-bls-GW, and pEU-FLAG-GST-MCS plasmids were constructed by polymerase chain reaction (PCR) using the In-Fusion system (Takara Bio) or PCR and restriction enzymes. The pEU-FLAG-GST-SALL4, -IKZF1 -DCAF1, -DCAF2, -ZNF536, -ZNF629, -BRD2, -BRD3 and -BRD4 plasmids were purchased from the Kazusa DNA Research Institute[65]. AirID was purchased as an artificial gene from Thermo Fisher Scientific, and point mutations were performed by inverse PCR. The open reading frames (ORFs) of *SALL4*, *IKZF1*, and *ZNF536* were amplified, and restriction enzyme sites were added by PCR and cloned into pcDNA3.1(+)-AGIA-MCS or pcDNA3.1(+)-Myc-MCS. The ORFs of *CRBN*, *ADNP*, *CSTF1*, *EEF1A1*, *GFPT1*, *NOSIP*, *ZNF687*, *FGFR1*, *ZMYM4*, and *DCAF15* were purchased from the Mammalian Gene Collection (MGC), while the ORFs of *ZMYM2 and ZMYM3* were purchased from Promega (Flexi Clone). The BP reaction sequence (attB and attP) was added to CRBN by PCR and cloned into pDONR221 using BP recombination (Invitrogen/Thermo

Fisher Scientific). Then, pDONR221-CRBN was recombined into pEU-bls-GW or pcDNA3.1(+)-FLAG-GW using LR recombination (attL and attR). The restriction enzyme sites were added to ADNP, CSTF1, EEF1A1, GFPT1, NOSIP, ZNF687, FGFR1, ZMYM2, ZMYM3, and ZMYM4 by PCR. The ORFs were cloned into pEU-FLAG-GST-MCS. ZNF687 was cloned into pcDNA3.1(+)-Myc-MCS. FGFR1 and ZMYM2 were cloned into pCAGGS-MCS-AGIA. ZMYM2-FGFR1 was constructed by PCR and the restriction enzyme XhoI and cloned into pEU-FLAG-GST-MCS or pCAGGS-MCS-AGIA. The truncated ZMYM2s were generated by PCR and cloned into pEU-FLAG-GST-MCS using restriction enzymes. ZMYM2-C470A and ZMYM2-FGFR1-C470A were mutated by inverse PCR.

For the generation of lentivirus for stable cell lines, AGIA-AirID-CRBN-WT, AGIA-AirID-CRBN-YW/AA, AGIA-BioID-CRBN-WT, AGIA-TurboID-CRBN-WT, ZMYM2-AGIA, FGFR1-AGIA, ZMYM2-FGFR1-WT-AGIA, ZMYM2-FGFR1-C470A-AGIA, AGIA-AirID-DCAF15-WT, and AGIA-AirID-DCAF15-D475N were cloned into the pCSII-CMV-MCS-IRES2-Bsd vector (RIKEN) using restriction enzymes.

**Cell culture and transfection**. HEK293T and HCT116 cells were cultured in low-glucose DMEM (FUJIFILM Wako Pure Chemical Corporation) supplemented with 10% foetal bovine serum (FUJIFILM Wako), 100 U/mL penicillin, and 100 μg/mL streptomycin (Gibco/Thermo Fisher Scientific) at 37 °C under 5% CO$_2$. HEK293T cells were transiently transfected using *Trans*IT-LT1 transfection reagent (Mirus Bio) or polyethyleneimine (PEI) Max (MW 40,000) (PolyScience, Inc.).

HuH7 cells were cultured in DMEM (high glucose) medium (FUJIFILM Wako) supplemented with 10% foetal bovine serum, 100 U/mL penicillin, 100 μg/mL streptomycin, 1 mM sodium pyruvate (Gibco), 10 mM HEPES (Gibco), and 1× MEM NEAA (Gibco) at 37 °C under 5% CO$_2$.

THP-1, MM1.S, and U266 cells were cultured in RPMI160 GlutaMAX medium (Gibco) supplemented with 10% foetal bovine serum (FUJIFILM Wako), 100 U/mL penicillin, 100 μg/mL streptomycin, and 55 μM 2-mercaptoethanol (Gibco) at 37 °C under 5% CO$_2$.

IMR32 cells were cultured in MEM GlutaMAX medium (Gibco) supplemented with 10% foetal bovine serum, 100 U/mL penicillin, 100 μg/mL streptomycin, and 1× MEM NEAA at 37 °C under 5% CO$_2$.

**Generation of stable cell lines using lentivirus**. Each lentivirus was produced in HEK293T cells by transfection of pCSII-CMV-ORF-IRES2-Bsd expression vector together with pCMV-VSV-G-RSV-Rev and pCAG-HIVgp. After 24 h of transfection, the culture medium was exchanged with fresh medium, and the cells were cultured for 48 h. Then, the lentiviruses were concentrated using a Lenti-X concentrator (Takara Bio). HEK293T, HuH7, or IMR32 cells supplemented with 10 μg/mL polybrene (Nacalai Tesque) were infected with the appropriate lentivirus. In the case of THP-1 or MM1.S, the cells were suspended in 50 μL culture medium containing 8 μg/mL polybrene and lentivirus and infected by incubation at 37 °C for 30 min. After 24 h of infection, the culture medium was exchanged, and 10 μg/ mL blasticidin S (InvivoGen) selection was started 24 h after the culture medium was exchanged.

**Antibodies**. The following horseradish peroxidase (HRP)-conjugated antibodies were used in this study: FLAG (Sigma-Aldrich, A8592, 1:5000), AGIA[27] (produced in our laboratory), Myc-tag (Cell Signaling Technology, #2040, 1:1000), α-tubulin (MBL, PM054-7, 1:5000), and biotin (Cell Signaling Technology, #7075, 1:3000). The following primary antibodies were used in this study: CRBN (#71810, 1:1000), IKZF1/Ikaros (#14859, 1:1000), IKZF3/Aiolos (#15103, 1:1000), GSPT1 (#14980, 1:1000), phospho-STAT1 (#9167, 1:1000), STAT1 (#9176, 1:1000), phospho-STAT3 (#9145, 1:1000), STAT3 (#9132, 1:1000), phospho-ERK1/2 (#4377, 1:1000) and ERK1/2 (#4695, 1:1000) (all from Cell Signaling Technology); ZNF687 (A303-278A, 1:1000), WIZ (A305-864A, 1:1000), ZFP91 (A303-245A, 1:1000), BRD4 (A301-985A, 1:1000), BRD2 (A302-583A, 1:1000) (all from Bethyl Laboratories); PLZF (R&D System, AF2944, 1:1000); SALL4 (Abcam, ab29112, 1:1000; SALL4 (Santa Cruz Biotechnology, sc-101147, 1:500); ZMYM2 (GeneTex, GTX105550, 1:1000); ZMYM2 (Gene Tex, GTX31821, 1:1000); CK1α (Abcam, ab108296, 1:1000); BRD3 (Proteintech, 11859-1-AP, 1:1000); RBM39 (Sigma-Aldrich, HPA001591, 1:1000); and α-tubulin (LI-COR Biosciences, 926-42213, 1:1000). Anti-rabbit IgG (HRP-conjugated, Cell Signaling Technology, #7074, 1:10,000), anti-mouse IgG (HRP-conjugated, Cell Signaling Technology, #7076, 1:10,000), anti-goat IgG (HRP-conjugated, Invitrogen/Thermo Fisher Scientific, #81-1620, 1:10,000), IRDye 800CW goat anti-rabbit IgG (LI-COR Biosciences, 925-32211, 1:10,000) and IRDye 680RD goat anti-mouse IgG (LI-COR Biosciences, 925-68070, 1:10,000) were used as secondary antibodies.

**Immunoblot analysis**. Protein samples were separated by SDS-PAGE and transferred to polyvinylidene difluoride (PVDF) membranes (Millipore). The membranes were blocked using 5% skim milk (Megmilk Snow Brand) in TBST (20 mM Tris-HCl [pH 7.5], 150 mM NaCl, 0.05% Tween20) at room temperature for 1 h, and then treated with the appropriate antibodies. Immobilon (Millipore), ImmunoStar LD (FUJIFILM Wako), or EzWestLumi plus (Atto) were used as substrates for HRP, and the luminescence signal was detected using an ImageQuant LAS

4000 mini (GE Healthcare). In some blots, the membrane was stripped with stripping solution (FUJIFILM Wako) and re-probed with other antibodies.

For fluorescent immunoblot analysis, the membranes were blocked using Intercept (TBS) Blocking Buffer (LI-COR Biosciences) at room temperature for 1 h, and then treated with the appropriate antibodies. The fluorescent signal was detected using an Odyssey Fc (LI-COR Biosciences). Relative ZMYM2 protein levels were analysed using Empiria Studio software (LI-COR Biosciences).

All immunoblot data were analysed using ImageJ software.

**Production of recombinant proteins using a cell-free system.** Recombinant protein synthesis was conducted using a wheat cell-free system. In vitro transcription and wheat cell-free protein synthesis were performed using the WEPRO1240 expression kit (Cell-Free Sciences). Transcription was performed using SP6 RNA polymerase with the plasmids or DNA fragments as templates. The translation reaction was performed in bilayer mode using the WEPRO1240 expression kit (Cell-Free Sciences), according to the manufacturer's protocol. For biotin labelling of bls-CRBN, cell-free synthesised crude biotin ligase (BirA) produced using the wheat cell-free expression system was added to the bottom layer, and 0.5 μM (final concentration) of d-biotin (Nacalai Tesque) was added to both the upper and lower layers, as described previously[66].

**In vitro biotinylation of neo-substrates and streptavidin pull-down assay.** Reaction mixtures (50 μL) containing 20 μL recombinant FLAG-GST-IKZF1 or -SALL4, 20 μL recombinant AGIA-AirID-CRBN, biotin (final conc. 500 nM), NaCl (final conc. 100 mM), ATP (final conc. 960 μM), and IMiDs (final conc. 20 μM/1% DMSO) were prepared. Subsequently, biotinylation reactions were performed by incubation at 26 °C for 3 h. Then, 410 μl of 50 mM Tris-HCl (pH 7.5) was added to 40 μL of each reaction mixture. After the addition of 50 μL of 10% SDS, the reaction mixtures were boiled at 95 °C for 15 min. Then, each mixture was diluted 3-fold with 1.5 mL IP Lysis buffer (Pierce) (25 mM Tris-HCl pH 7.5, 150 mM NaCl, 1 mM EDTA, 1% NP-40, 5% glycerol). The biotinylated proteins were captured using 10 μL streptavidin Sepharose beads (GE Healthcare) and the mixtures were rotated at 26 °C for 2 h. The beads were washed three times with 800 μL IP lysis buffer and the proteins were eluted with 40 μL of 1× sample buffer (62.5 mM Tris-HCl pH 6.8, 2% SDS, 10% glycerol).

**Streptavidin pull-down assay (STA-PDA) of neo-substrates in cells.** CRBN$^{−/−}$ HEK293T cells were cultured in 6-well plates and transfected with 500 ng pcDNA3.1(+)-AGIA-AirID-CRBN-WT or 800 ng pcDNA3.1(+)-AGIA-AirID-CRBN-YW/AA together with 500 ng pcDNA3.1(+)-Myc-SALL4. After the cells were transfected for 24 h, they were treated with 10 μM pomalidomide or DMSO in the presence of 10 μM biotin and 5 μM MG132 for 6 h. The cells were harvested by suspension in TrypLE Select (Gibco). The cell pellets were washed with 1× PBS and lysed in 150 μL SDS lysis buffer (50 mM Tris-HCl pH 7.5, 2% SDS) containing a protease inhibitor cocktail (Sigma-Aldrich), and the lysates were denatured by boiling at 95 °C for 15 min. The lysates were then diluted 2-fold with 150 μL of 50 mM Tris-HCl pH 7.5, and clarified by centrifugation at 16,100 × g for 15 min. Then, 560 μL of lysate was added to 1 mL IP lysis buffer containing 30 μL streptavidin Sepharose beads. After rotating at 4 °C overnight, the beads were washed three times with 800 μL IP lysis buffer, and the proteins were eluted by boiling with 50 μL of 1× sample buffer containing 5% 2-mercaptoethanol.

For the streptavidin pull-down assay of exogenous neo-substrates in HEK293T cells stably expressing AGIA-AirID-CRBN-WT or AGIA-AirID-CRBN-YW/AA, the cells were cultured in 6-well plates and transfected with 500 ng pcDNA3.1(+)-Myc-SALL4 and 500 ng pcDNA3.1(+)-Myc-IKZF1. After the cells were transfected for 24 h, the cells were treated with 10 μM pomalidomide or DMSO in the presence of 10 μM biotin and 10 μM MG132 for 6 h. The cells were harvested by suspension in TrypLE Select (Gibco). The cell pellets were washed with 1× PBS and lysed in 150 μL SDS lysis buffer (50 mM Tris-HCl pH 7.5, 2% SDS) containing a protease inhibitor cocktail (Sigma-Aldrich), and the lysates were denatured by boiling at 95 °C for 15 min. The lysates were then diluted 2-fold with 150 μL of 50 mM Tris-HCl pH 7.5 and clarified by centrifugation at 16,100 × g for 15 min. Then, 560 μL of lysate was added to 1 mL IP lysis buffer containing 30 μL streptavidin Sepharose beads. After rotating at 4 °C overnight, the beads were washed three times with 800 μL IP lysis buffer, and the proteins were eluted by boiling with 50 μL of 1× sample buffer containing 5% 2-mercaptoethanol.

To prepare the streptavidin pull-down assay using AirID-CRBN-expressing cells, THP-1, HEK293T, MM1.S, HuH7, and IMR32 cells stably expressing AGIA-AirID-CRBN-WT or -YW/AA were cultured in a 10-cm dish and treated with DMSO, IMiDs, dBET1, or ARV-825 in the presence of biotin and MG132 at the indicated times and concentrations. The cells were harvested by suspension in TrypLE Select (Gibco). The cell pellets were washed with 1× PBS and lysed in 300 μL SDS lysis buffer (50 mM Tris-HCl pH 7.5, 2% SDS) containing a protease inhibitor cocktail (Sigma-Aldrich), and the lysates were denatured by boiling at 95 °C for 15 min. Then, the lysates were diluted 2-fold with 300 μL of 50 mM Tris-HCl pH 7.5 and clarified by centrifugation at 16,100 × g for 15 min. Then, 560 μL of lysate was added to 1 mL IP lysis buffer containing 30 μL streptavidin Sepharose beads. After rotating at 4 °C overnight, the beads were washed three times with

800 μL IP lysis buffer, and the proteins were eluted by boiling with 45 μL of 1× sample buffer containing 5% 2-mercaptoethanol.

To compare BioID-CRBN, AirID-CRBN, and TurboID-CRBN, HEK293T, or IMR32 cells stably expressing AGIA-BioID-CRBN, cultures of AGIA-AirID-CRBN or AGIA-TurboID-CRBN were prepared in a 10 cm dish. The cells were treated for 2 h or 6 h with DMSO or 10 μM pomalidomide in the presence of biotin and 5 μM MG132 and harvested by suspension in TrypLE Select (Gibco). The cell pellets were washed with 1× PBS and lysed in 300 μL SDS lysis buffer (50 mM Tris-HCl pH 7.5, 2% SDS) containing a protease inhibitor cocktail (Sigma-Aldrich), and the lysates were denatured by boiling at 95 °C for 15 min. The lysates were then diluted 2-fold with 300 μL of 50 mM Tris-HCl pH 7.5, and clarified by centrifugation at 16,100 × g for 15 min. Then, 560 μL of lysate was added to 1 mL IP lysis buffer containing 30 μL streptavidin Sepharose beads. After rotating at 4 °C overnight, the beads were washed three times with 800 μL IP lysis buffer, and the proteins were eluted by boiling with 45 μL of 1× sample buffer containing 5% 2-mercaptoethanol.

For preparation of the streptavidin pull-down assay using AirID-DCAF15-expressing cells, HCT116 cells stably expressing AGIA-AirID-DCAF15-WT or -D475N were cultured in a 10-cm dish and treated with DMSO or 5 μM indisulam in the presence of 50 μM biotin and 10 μM MG132 for 6 h. The cells were harvested by suspension in TrypLE Select (Gibco). The cell pellets were washed with 1× PBS and lysed in 250 μL RIPA buffer (25 mM Tris-HCl pH 8.0, 150 mM NaCl, 1% NP-40, 0.5% sodium deoxycholate, 0.1% SDS, 1 mM EDTA) containing a protease inhibitor cocktail (Sigma-Aldrich), followed by sonication for 1 min. In addition, the lysates were clarified by centrifugation at 21,500 × g for 10 min, and then denatured by incubation at 37 °C for 30 min in the presence of 1% SDS. The lysates were added to 20 μL IP wash buffer (50 mM Tris-HCl pH 7.5, 1% TritonX-100, 150 mM NaCl, and 0.5 mM EDTA) containing streptavidin Sepharose beads. After rotation at room temperature for 1 h, the beads were washed three times with 500 μL IP wash buffer, and the proteins were eluted by boiling with 30 μL 2× sample buffer containing 5% 2-mercaptoethanol.

**Comparison of biotinylated proteins between cell lines.** For comparison of biotinylation levels in cells, HEK293T, MM1.S, HuH7, and IMR32 cells stably expressing AGIA-AirID-CRBN-WT were cultured in 12-well plates and treated with DMSO, 10 μM thalidomide, or 10 μM pomalidomide in the presence of 10 μM biotin and 5 μM MG-132 for 8 h. MM1.S cells were harvested by suspension and centrifuged at 400 × g for 3 min, after which the cell pellets were washed with PBS. In the case of HEK293T, HuH7, or IMR32 cells, the cells were harvested using TrypLE Select and centrifuged at 400 × g for 3 min, and the cell pellets were washed with PBS. Then, the cell pellets were lysed with 150 μL RIPA buffer containing a protease inhibitor cocktail by vortexing and incubation at 4 °C for 15 min. The cell lysates were clarified by centrifugation at 16,100 × g for 15 min and protein concentrations of the lysates were quantified by BCA protein assay (Invitrogen). Then, the lysates of the same protein level were denatured in 1× sample buffer containing 5% 2-mercaptoethanol by boiling.

**Preparation of cell lysates treated with IMiDs, indisulam, or PROTACs for enrichment of biotinylated peptides.** For preparation of IMiDs-treated THP-1 or HEK293T cells stably expressing AGIA-AirID-CRBN-WT, the cells were cultured in a 10-cm dish in biological replicates (n = 3) and treated with DMSO or the indicated concentrations of IMiDs in the presence of 10 μM biotin and 5 μM MG132 for 8 h. The cells were harvested by suspension in TrypLE Select (Gibco) and the cell pellets were washed with PBS. The cells were then lysed in 250 μL (THP-1 cells) or 300 μL (HEK293T cells) Gdm-TCEP buffer (6 M guanidine-HCl, 100 mM HEPES-NaOH pH 7.5, 10 mM TCEP, 40 mM chloroacetamide).

For the IMiDs-treated MM1.S, HuH7, or IMR32 cells stably expressing AGIA-AirID-CRBN-WT, the cells were cultured in a 10-cm dish in biological replicates (n = 3) and treated with DMSO or the indicated concentrations of IMiDs in the presence of 10 μM biotin and 5 μM MG132 for 8 h. The cells were harvested by suspension in TrypLE Select (Gibco) and pooled in one tube, after which the cell pellets were washed with PBS. The cells were then separated into three tubes and lysed in 300 μL (MM1.S cells) or 400 μL (HuH7 and IMR32 cells) of Gdm-TCEP buffer (6 M guanidine-HCl, 100 mM HEPES-NaOH pH 7.5, 10 mM TCEP, and 40 mM chloroacetamide).

To compare the enrichment methods of biotinylated proteins or peptides, MM1.S cells stably expressing AGIA-AirID-CRBN-WT were cultured in a 10 cm dish in three biological replicates and treated with DMSO or 10 μM pomalidomide in the presence of 10 μM biotin and 5 μM MG132 for 8 h. The cells were harvested by suspension in TrypLE Select (Gibco) and pooled in one tube, after which the cell pellets were washed with PBS. The cells were then separated into three tubes and lysed in 300 μL (MM1.S cells) of Gdm-TCEP buffer (6 M guanidine-HCl, 100 mM HEPES-NaOH pH 7.5, 10 mM TCEP, and 40 mM chloroacetamide).

To compare AirID-CRBN and TurboID-CRBN, IMR32 cells stably expressing AGIA-AirID-CRBN or AGIA-TurboID-CRBN were cultured in a 10 cm dish in three biological replicates and treated with DMSO or 10 μM pomalidomide in the presence of 10 μM biotin and 5 μM MG132 for 2 h. The cells were harvested by suspension in TrypLE Select (Gibco) and pooled in one tube, after which the cell pellets were washed with PBS. The cells were then separated into three tubes and lysed in 300 μL of Gdm-TCEP buffer.

To prepare PROTACs-treated MM1.S cells stably expressing AGIA-AirID-CRBN-WT, the cells were cultured in a 10-cm dish and treated with DMSO, 1 μM pomalidomide, or 0.1 μM ARV-825 in the presence of 5 μM biotin and 5 μM MG132 for 6 h. The cells were harvested by suspending and pooled in one tube, and the cell pellets were washed with PBS. Then, the cells were separated into three tubes and lysed in 300 μL Gdm-TCEP buffer.

To prepare indisulam-treated HCT116 cells stably expressing AGIA-AirID-CRBN-WT or -D475N, the cells were cultured in a 6-cm dish and treated with DMSO or 5 μM indisulam in the presence of 50 μM biotin and 10 μM MG132 for 6 h. The cells were harvested using TrypLE Select and pooled in one tube, after which the cell pellets were washed with PBS. The cells were separated into three tubes and lysed in 200 μL Gdm-TCEP buffer.

**Enrichment of biotinylated peptides using tamavidin 2-REV or anti-biotin antibody**. The cell lysates in Gdm-TCEP buffer were dissolved by heating and sonication and then centrifuged at $20,000 \times g$ for 15 min at 4 °C. The supernatants were recovered, and proteins were purified by methanol–chloroform precipitation and solubilised using 0.1% RapiGest SF (Waters) in 50 mM triethylammonium bicarbonate. After repeated sonication and vortexing, the proteins were digested with trypsin (MS grade, Thermo Fisher Scientific) at 37 °C overnight. The resultant peptide solutions were diluted 5-fold with TBS (50 mM Tris-HCl pH 7.5, 150 mM NaCl). Biotinylated peptides were captured on 15 μL slurry of MagCapture HP Tamavidin 2-REV magnetic beads (FUJIFILM Wako) by incubation for 3 h at 4 °C. After washing with TBS five times, the biotinylated peptides were eluted with 100 μL of 1 mM biotin in TBS for 15 min at 95 °C twice. Alternatively, biotinylated peptides were captured on a 20 μL slurry of the anti-biotin (A7C2A) rabbit mAb–conjugated beads (Cell Signaling Technology) by incubation for 3 h at 4 °C. After washing with TBS four times and with ultrapure water twice, the biotinylated peptides were eluted with 100 μL of 0.2% TFA in 80% ACN for 10 min at room temperature twice. The combined eluates were desalted using GL-Tip SDB (GL Sciences), evaporated in a SpeedVac concentrator (Thermo Fisher Scientific), and redissolved in 0.1% TFA and 3% acetonitrile (ACN).

**Enrichment of biotinylated proteins using streptavidin beads for LC-MS/MS analysis**. Proteins in the cell lysates in Gdm-TCEP buffer were purified by methanol–chloroform precipitation as described above and solubilised using 8 M urea and 1% SDS in TBS. After sonication and 4-fold dilution with TBS, biotinylated proteins were captured on a 10 μL slurry of NanoLink Streptavidin magnetic beads (Solulink) by incubation for 3 h at 4 °C. After washing with 2 M urea and 0.25% SDS in TBS four times and with 1 M urea in 50 mM ammonium bicarbonate three times, proteins on the beads were digested by adding 200 ng trypsin/Lys-C mix (Promega) at 37 °C overnight. The digests were acidified, desalted using GL-Tip SDB, evaporated, and dissolved in 0.1% TFA and 3% acetonitrile (ACN).

**Data-dependent LC-MS/MS analysis**. LC-MS/MS analysis of the resultant peptides was performed on an EASY-nLC 1200 UHPLC connected to an Orbitrap Fusion mass spectrometer through a nanoelectrospray ion source (Thermo Fisher Scientific). The peptides were separated on a 150-mm $C_{18}$ reversed-phase column with an inner diameter of 75 μm (Nikkyo Technos) with a linear 4–32% ACN gradient for 0–60 min, followed by an increase to 80% ACN for 10 min. The mass spectrometer was operated in data-dependent acquisition mode with a maximum duty cycle of 3 s. MS1 spectra were measured with a resolution of 120,000, an automatic gain control (AGC) target of $4 \times 10^5$, and a mass range of 375–1500 $m/z$. HCD MS/MS spectra were acquired in the linear ion trap with an AGC target of $1 \times 10^4$, an isolation window of 1.6 $m/z$, a maximum injection time of 200 ms, and a normalised collision energy of 30. Dynamic exclusion was set to 10 s. Raw data were directly analysed against the Swiss-Prot database restricted to *Homo sapiens* using Proteome Discoverer version 2.4 (Thermo Fisher Scientific) with the Mascot search engine. The search parameters were as follows: (a) trypsin as an enzyme with up to two missed cleavages; (b) precursor mass tolerance of 10 ppm; (c) fragment mass tolerance of 0.6 Da; (d) carbamidomethylation of cysteine as a fixed modification; and (e) acetylation of protein N terminus, oxidation of methionine, and biotinylation of lysine as variable modifications. Peptides were filtered at a false discovery rate (FDR) of 1% using the Percolator node. Label-free quantification was performed based on the intensities of precursor ions using the precursor ion quantifier node. Normalisation was performed such that the total sum of abundance values for each sample over all peptides was the same. For statistical analyses of MS data, the *P*-values in each volcano plot were calculated using Student's two-sided *t*-tests. The adjusted *P*-values were calculated by controlling the FDR and are shown in the Supplementary Data.

**AlphaScreen-based biochemical assays using recombinant proteins**. Biotinylated bls-CRBN mixtures (0.5 μL) were prepared in 10 μL AlphaScreen buffer containing 100 mM Tris (pH 8.0), 0.01% Tween20, 100 mM NaCl, and 1 mg/mL BSA. IMiDs or PROTACs mixtures at the concentrations indicated were prepared in 5 μL AlphaScreen buffer. Substrate mixtures (5 μL) containing 0.8 μL FLAG-GST-neo-substrate in AlphaScreen buffer were prepared. Subsequently, the three mixtures were mixed and incubated at 26 °C for 1 h in a 384-well AlphaPlate

(PerkinElmer). Then, 5 μL detection mixture containing 0.2 μg/mL anti-DYKDDDDK mouse mAb (Wako), 0.08 μL streptavidin-coated donor beads, and 0.08 μL Protein A-coated acceptor beads (μL) in AlphaScreen buffer were added to each well. After incubation at 26 °C for 1 h, luminescence signals were detected using an EnVision plate reader (PerkinElmer).

**IMiDs- or PROTACs-dependent protein degradation assays in cells**. To confirm the protein degradation of neo-substrates in stable cell lines expressing AirID-CRBN, HEK293T or HuH7 cells were cultured in 24-well plates and treated with 10 μM pomalidomide or DMSO (0.1%) and biotin in culture medium at the indicated times and concentrations.

To confirm IMiDs-dependent degradation of exogenous ZNF536, ZNF687, and ZMYM2, HEK293T cells were cultured in 24-well plates and transfected with 300 ng pcDNA3.1(+)-FLAG-CRBN-WT together with 50 ng pcDNA3.1(+)-AGIA-ZNF536, 200 ng pcDNA3.1(+)-ZNF687-Myc, or 200 ng pCAGGS-ZMYM2-AGIA. After the cells were transfected for 6 h, they were treated with IMiDs or DMSO (0.1%) in culture medium at the indicated times and concentrations.

To degrade ZMYM2, FGFR1, or ZMYM2-FGFR1, HEK293T cells were cultured in 24-well plates and transfected with 400 ng pcDNA3.1(+)-FLAG-CRBN-WT together with 200 ng pCAGGS-ZMYM2-AGIA, 20 ng pCAGGS-FGFR1-AGIA, or 40 ng pCAGGS-ZMYM2-FGFR1-AGIA. After the cells were transfected for 6 h, they were treated with IMiDs or DMSO (0.1%) in culture medium at the indicated times and concentrations.

To examine the selectivity for protein degradation of ZMYM2 and ZMYM2-FGFR1, HEK293T cells were cultured in 24-well plates and transfected with 300 ng pcDNA3.1(+)-FLAG-CRBN-WT together with 200 ng pCAGGS-ZMYM2-AGIA or 10 ng pCAGGS-. After the cells were transfected for 6 h, they were treated with IMiDs or DMSO (0.1%) in culture medium at the indicated times and concentrations.

For degradation of ZMYM2 cells, HEK293T cells were cultured in 24-well plates and transfected with 300 ng pcDNA3.1(+)-FLAG-CRBN-WT together with 200 ng pCAGGS-ZMYM2-WT-AGIA or 200 ng pCAGGS-ZMYM2-C470A-AGIA. After the cells were transfected for 6 h, they were treated with IMiDs or DMSO (0.1%) in culture medium at the indicated times and concentrations.

To examine the selectivity of IMiDs in multiple-myeloma cell lines, MM1.S cells or U266 cells were cultured in a 24-well plate and treated with IMiDs or DMSO (0.1%) for 24 h at the indicated concentrations.

To examine the degradation of endogenous GSPT1, ZNF687, ZMYM2, HEK293T, HuH7, or IMR32 cells were cultured in 24-well plates. Then, the cells were treated with IMiDs or DMSO (0.1%) in culture medium for 24 h at the indicated concentrations.

To show that pomalidomide-dependent ZMYM2 degradation is caused by the Cullin E3 ligase complex and the 26S proteasome, IMR32 or HEK293T cells were cultured in 12-well plates and treated with DMSO, 1 μM pomalidomide, or 10 μM pomalidomide in the presence of 1 μM MLN4924 or 5 μM MG132 (0.2% DMSO) for 24 h.

For fluorescent immunoblot analysis of protein degradation of ZMYM2, IMR32 cells were cultured in 24-well plates. The cells were then treated with pomalidomide or DMSO (0.1%) in culture medium for 48 h at the indicated concentrations.

To degrade ZMYM2-FGFR1 in stable cell lines, HEK293T cells expressing empty, ZMYM2-AGIA, FGFR1-AGIA, or ZMYM2-FGFR1-AGIA were cultured in 24-well plates. The cells were then treated with IMiDs or DMSO (0.1%) in culture medium at the indicated times and concentrations.

For PROTACs-dependent degradation of BRD proteins and neo-substrates, MM1.S cells or HuH7 cells were cultured in 24-well plates and treated with dBET1, ARV-825, IMiDs, or DMSO (0.1%) for 24 h, at the indicated concentrations.

In all experiments, cells were lysed by boiling in 1× sample buffer containing 5% 2-mercaptoethanol, and the lysates were analysed by immunoblotting.

**Quantitative RT-PCR**. To show that pomalidomide-dependent protein degradation of ZMYM2 is downregulated at the post-translational level, ZMYM2 mRNA expression in HEK293T or IMR32 cells treated with pomalidomide or DMSO (0.1%) for 24 h was examined by qRT-PCR. Total RNA was isolated from the cells using the SuperPrep Cell Lysis Kit (Toyobo) and cDNA was synthesised using the SuperPrep RT kit (Toyobo), according to the manufacturer's protocol. RT-PCR was performed using KOD SYBR qPCR Mix (Toyobo) and the data were normalised against GAPDH mRNA levels. PCR primers were as follows: ZMYM2 sense 5′-CTAACTGAGATTCGCCATGAAGTC-3′, ZMYM2 antisense 5′-CTCT CCACACTGTTCACAGCAATTC-3′, GAPDH sense 5′-AGCAACAGGG TGGTGGAC-3′, and GAPDH antisense 5′-GTGTGGTGGGGGACTGAG-3′.

**Cell viability assay in MM1.S or U266 cells**. MM1.S cells and U266 cells were seeded at $6 \times 10^5$ cells/well and cultured in 12-well plates in the presence of IMiDs, PROTACs, or DMSO (0.1%) for 3 days at the indicated concentrations. Then, the cells were diluted twice by centrifugation and cultured in the presence of the same IMiDs, PROTACs, or DMSO (0.1%) for an additional 3 days. Viable cells were counted using the CellTiter-Glo assay kit (Promega).

**Statistical analysis and reproducibility**. The data are presented as the mean ± standard deviation (SD). Significant changes were analysed by Student's t-tests using Microsoft Excel spreadsheets with basic statistical program or one-way analysis of variance (ANOVA), followed by Tukey's test using GraphPad Prism (version 8) software (GraphPad, Inc.). Immunoblot analyses and streptavidin pull-down assays were repeated more than twice with similar results.

**Reporting summary**. Further information on research design is available in the Nature Research Reporting Summary linked to this article.

## Data availability

The MS proteomics data have been provided in Supplementary Data 1–12 and deposited to the ProteomeXchange Consortium via the jPOST partner repository with the dataset identifiers PXD028754 (three enrichment methods using AirID-CRBN-expressing MM1.S cells), PXD028755 (neo-substrate selectivity of IMiDs using AirID-CRBN-expressing MM1.S cells), PXD028756 (LC-MS/MS analysis of IMiDs-dependent biotinylated peptides using AirID-CRBN-expressing HEK293T cells), PXD028757 (LC-MS/MS analysis of IMiDs-dependent biotinylated peptides using AirID-CRBN-expressing HuH7 cells), PXD028758 (LC-MS/MS analysis of IMiDs-dependent biotinylated peptides using AirID-CRBN-expressing IMR32 cells), PXD028760 (LC-MS/MS analysis of IMiDs-dependent biotinylated peptides using AirID-CRBN-expressing THP-1 cells), PXD028761 (Validation of AirID and TurboID using AirID-CRBN- or TurboID-CRBN-expressing IMR32 cells), PXD028762 (LC-MS/MS analysis of Indisulam-dependent biotinylated peptides using AirID-DCAF15-expressing HCT116 cells) and PXD028763 (LC-MS/MS analysis of PROTACs-dependent biotinylated peptides using AirID-CRBN-expressing MM1.S cells). All data supporting the findings of this study are provided in the main text and Supplementary information. Source data are provided with this paper.

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

## Acknowledgements

We thank Y. Shoya and M. Kawano for technical assistance, and the Applied Protein Research Laboratory of Ehime University. We also thank Prof. T. Okamoto (Osaka University) for providing the HuH7 cell line. This work was mainly supported by the Platform Project for Supporting Drug Discovery and Life Science Research (Basis for Supporting Innovative Drug Discovery and Life Science Research (BINDS)) from the Japan Agency for Medical Research and Development (AMED) under Grant Number JP21am0101077 (T.S.), the Project for Cancer Research and Therapeutic Evolution (P-CREATE) grant number (JP21cm0106181h0006 for S.Y.) from AMED, a Grant-in-Aid for Scientific Research on Innovative Areas (21H00285 for S.Y., JP16H06579 for T.S. and JP19H04966 for H.K) from the Japan Society for the Promotion of Science (JSPS). This work was also partially supported by JSPS KAKENHI (21K15076 for S.Y., JP19H03218 for T.S., and JP17H06112 for N.S.), Takeda Science Foundation, and Joint Usage and Joint Research Programs of the Institute of Advanced Medical Sciences, Tokushima University.

## Author contributions

S.Y. and Y.H. performed the biochemical, molecular, and cellular biology experiments of thalidomide and its derivatives. S.Y. and S.M. performed the biochemical, molecular, and cellular biology experiments of PROTACs. K.K. performed cellular biology experiments of indisulam. M.M. and N.S. synthesised and analysed the 5-hydroxythalidomide. K.N. and H.K. performed enrichment of biotinylated peptides and LC-MS/MS analyses. S.Y. and T.S. analysed the data, designed the study, wrote the paper, and all authors contributed to the manuscript.

## Competing interests

The authors declare no competing interests.
