## [Peer Review File · Nature Communications]

REVIEWER COMMENTS

Reviewer #1 (Remarks to the Author):

This paper presents additional applications of the previously reported AirID-CRBN fusion to identify neosubstrates of CRBN with thalidomide and derivatives and PROTACs and extension to AirID-DCAF15 and the molecular glue indisulam. Although the AirID-CRBN system was briefly presented in a prior publication, showing that known neosubstrates are recruited to cereblon upon pomalidomide treatment [K. Kido et al., AirID, a novel proximity biotinylation enzyme, for analysis of protein-protein interactions. *Elife* 9, (2020)], this paper presents substantial new data, including discovery of a novel CRBN immunomodulatory drug-inducible neosubstrate with potential clinical significance, application to profiling PROTACs, and application to DCAF15. This paper would be a good fit for publication in Nature Communications with the following revisions addressed.

Reviewer suggestions:

- Use of biotin site identification for quantification gives higher resolution, but is technically more challenging given the necessity of label free quantification and may be a barrier for use of this method by others. Do the authors see similar quantification values when evaluating these data by quantitative proteomics?
- Presumably the identification of the novel zinc finger proteins in these cell lines by AirID-CRBN means that they are natively expressed in cells to a significant enough degree that the native protein would be able to be detected and validated by Western blot rather than expression of a fusion protein. A Western blot validating at least one or two of these targets in the appropriate cell line, particularly of the zinc finger of focus, ZMYM2, would support the authors conclusions about these proteins as novel targets.
- In line with the above, the authors make an interesting but speculative connection of ZMYM2-FGFR1 to AML, but demonstration of this protein as a degraded target in an AML cell line would support. Pomalidomide does not have inherent activity against a number of AML cell lines, and statements in the discussion about the clinical relevance of these findings should be tempered.
- It is not clear why the non-PROTACs AlphaScreen data were performed at a single protein concentration. Optimization to justify the single concentration used should be included in the supplemental information.
- The authors state that “this method can identify degradation-independent interactors,” but this appears to be speculative with the current manuscript. The data indicate that this may be the case with ZNF536 and/or ZNF687, although additional validation would be necessary. Comparison of these zinc finger hits back to the consensus sequence that was identified in Ebert and co-workers, *Science*, 2018 would be helpful context for whether or not the AirID system is yielding an expected binding event or a novel binding event (as with ZMYM2).

- Figure 1C. WT and YW/AA CRBN expression are very uneven and may be the underlying reason for the difference in biotinylation of SALL4.
- Figure 1E and others, there are blots (particularly tubulin enriched) that are so light it is unclear if anything is shown. Please put a border around the blots or adjust the greyscale so the contrast is clearer.
- Several suggestions for clarification of the text in the manuscript are provided below:
 - o “by infection of lentivirus with HEK293T-CRBN-/- cells”, please rephrase to “by infection of HEK293T-CRBN-/- cells with lentivirus” (89-90)
 - o “which is an unbound form with CRBN” (273), please rephrase to “which is a form unbound by CRBN”
 - o “lenalidomide and pomalidomide are the main drugs for treating hematologic cancer, including multiple myeloma (MM)” (38-39) should say some types of hematologic cancer
 - o “A small molecule in cells can reportedly regulate biological functions via the induction of PPIs” (47-48) Please remove “reportedly”
 - o “Indisulam is a second molecular glue in mammalian cells” (285) – indisulam defines a second class of immunomodulatory molecular glues, but is not a second molecular glue.
 - o “Immunomodulatory drugs (IMiDs) such as thalidomide, lenalidomide, and pomalidomide have been the most characterised molecular glues” (331-332) – this is arguable relative to molecular glues like rapamycin and cyclosporin.
 - o “it is possible that there are unknown neo-substrates containing other types of ZNFs, although many analyses are required to identify such neo-substrates (354-355) – seems as though using this technique will mitigate these challenges.
 - o “bitotinylated” (63) - typo
 - o Kd ~ 10-15 (81) - needs units
 - o “the full length form” (143) - of what?
 - o “however, these phosphorylation levels were almost the same in both cell lines” (277-278) with or without drug treatment or both?
 - o Unclear what “high drug action” means (used several times in PROTAC section)

Reviewer #2 (Remarks to the Author):

In their manuscript, Yamanaka et al. performed comprehensive profiling of ubiquitin ligase interactors and drug-induced neo-substrates using proximity biotinylation. To this end, the newly developed ancestral BirA (AirID) which was described by the group in a previous publication (Kido et al., *elife* 2020) was fused to CRBN and other ubiquitin ligases, expressed in various cell lines, treated with drugs and subsequently biotin precipitation followed by mass spectrometry was performed to identify proteins in proximity of the ubiquitin ligase. Furthermore, the authors demonstrate that AirID technology can be successfully extrapolated to analyze PROTAC-based protein interactions. These analyses identified the major known neo-substrates for molecular glues including IKZF1/3, CK1A for lenalidomide, RBM39 for indisulam, and BRD4 for a BRD4-PROTAC, demonstrating the applicability and great potential of this approach that can overcome the low sensitivity of conventional co-immunoprecipitation that is hampered by weak enzyme-substrate interaction. ZMYM2, a potential target in cancer, was identified as a novel neo-substrate for pomalidomide/CRBN and subsequent analyses found that ZMYM2 interacts via its MYM-type zinc finger domain in contrast to C2H2-ZNF present in other thalidomide neo-substrates. The experiments are very well performed and of high technical quality. The work provides a systematic and comprehensive overview of drug-dependent interactions of CRBN and other ligases. However, the AirID method has been described previously including pomalidomide/CRBN as an example by the group (Kido et al., *elife* 2020), and most of the neo-substrates/ interactors were described before in several publications. So to me the major novel biological finding of this work is the identification of ZMYM2 as a pomalidomide-specific substrate that may be translated to a clinical application in cancer but unfortunately this is not further followed up in the current paper. In a very recent publication in *Blood Cancer Discovery* by Renneville et al. (<https://bloodcancerdiscov.aacrjournals.org/content/2/3/250>, not yet on Pubmed) identified ZMYM2 as a novel avadomide-dependent CRBN substrate. They demonstrated that avadomide is more potent than pomalidomide in degrading ZMYM2. This finding was corroborated with in vitro and in vivo analysis including human patient samples. Although this supports the use of ZMYM2 as target in cancer, it also takes some of the novelty of the current work.

Specific remarks:

-Besides drug-induced neo-substrates, does AirID also detect other interacting proteins like E3 ligase complex members (DDB1, CUL4 etc.) or E2s in the absence of drugs? This is also of interest to the field and should be reported.

-There are several described natural substrates of CRBN like MEIS2, MCT-1, CD98, and GL2 that in part are also influenced by thalidomide analogs. Have these proteins been detected in treated/ untreated cells?

-Is there a correlation between the degree of biotinylation of a neo-substrate and its ubiquitination/degradation?

-Neo-substrate biotinylation differs between cell lines. Could this be due to different levels of expression of the BirA-CRBN in these cells? What other reasons could explain this?

-Several ZN finger proteins like ZNF536, 687, and GSPT1 were found biotinylated by Airid-CRBN in the presence of lenalidomide and pomalidomide but no or only weak degradation is observed. How is this explained?

- The text and language needs to be substantially revised. Some sentences are illogical or hard to understand and there are many grammatical errors.

Example:

“In the case of the pan-ThD neo-substrate PLZF11, biotin supplementation increased its biotinylation in HEK293T cells (Extended Data Fig. 1c). However, biotinylated PLZF was not detected upon supplementation with the highest biotin concentration (500 μ M).”

-ThD is not a commonly used abbreviation, I would either spell out or use IMiD for immunomodulatory drug

Reviewer #3 (Remarks to the Author):

In their study Yamanaka et al. claim to have developed a new method for drug-inducible protein proximity analysis using a previously reported (Kido et al, eLIFE 2020) further developed version of the BioID biotin ligase, named AirID. The authors first introduce the method that allows to identify proteins in proximity to molecular glues (here IMiDs, indisulam) or PROTAC drugs (dBET1/ARV-825 for BRD4) by ectopic expression of a fusion protein of a drug-binding E3 ligase (CRBN, DCAF15) and the AirID biotin ligase for proximity labeling. The method robustly works in living cells across multiple cell lines and allows to identify well known neo-substrates of lenalidomide-bound CRBN, such as IKZF1 or IKZF3, with biotin WB and mass spectrometry as a read out. Of particular interest are the identification of ZMYM2 and its FGFR1 fusion protein ZMYM2-FGFR1 as novel neo-substrates of CRBN, as demonstrated by AirID proximity labeling, Alpha-Screen binding assays and neo-substrate degradation assays in living cells. ZMYM2-FGFR1 is an oncogene in very rare cases of acute myeloid leukemia (AML). Consequences on oncogenic signal transduction after ZMYM2-FGFR1 degradation via pomalidomide are demonstrated in HEK293 cells, but not in an AML relevant model, patient cells or an animal model.

The study is technically sound and the manuscript is mostly well written.

Major points of criticism:

- The title is misleading – “Drug-inducible protein proximity analysis using AirID” – sounds like a small molecule activates a BioID enzyme, which is not the case. The small molecule rather changes the affinity of the E3 ligase for neo-substrates

- The authors claim that their drug-induced BioID approach for molecular glues is a new technical approach. The methodological advancement is incremental though as compared to their earlier report (Kido et al, eLIFE 2020) and a very similar approach has been published previously by Pech et al (eLIFE 2019; <https://pubmed.ncbi.nlm.nih.gov/31452512/>). Unfortunately, the authors do not at all cite this study, although Pech et al have used a BioID approach to confirm RBM39-degradation via DCAF15 after indisulam treatment: “DCAF15-BioID cells were pre-treated with indisulam for 72 hr prior to biotin and MG132 addition. This treatment regime lead to substantial indisulam-dependent biotinylation of RBM39”. This is very similar to the author’s Figure 6 c in Yamanaka et al.

- Page 2, abstract: “Here, we demonstrate a new approach using AirID, a proximity biotinylation enzyme, to analyse target proteins proximately interacting with drugs in cells, such as molecular glues and PROTACs.”

The approach is not entirely new, see Pech et al, eLIFE 2019

- From a technology standpoint it is a major limitation of this and also the original AirID study, that the AirID biotin ligase is not properly benchmarked against the widely used TurboID (and also BirA*) biotin ligase in proximity proteomics experiments in living cells. The authors obviously propagate their AirID version of the BioID enzyme, but is it really better than TurboID for assays in living cells? The authors should show a proper comparison in living cells at low biotin concentrations and short labeling times for AirID vs TurboID with quantitative mass spectrometry as a readout (use pomalidomide/CRBN in IMR32). Hundreds of laboratories use BirA* or TurboID, why switch to AirID?

- With respect to AirID proximity proteomics it is also unclear why the authors only use the tamavidin 2-REV enrichment approach. Alternatives are capture of biotinylated peptides by widely available antibodies (as in Udeshi et al, Nat. Methods 2017; <https://pubmed.ncbi.nlm.nih.gov/29039416/>) or streptavidin pull downs with harsh washing conditions and on bead digest, which provide a lot better sensitivity but less selectivity as compared to the biotinylated peptide captures. The study would benefit a lot from a technological perspective if these approaches were compared in an application for identifying neo-substrates of CRBN.

- Figure 4e: Quantify effect of pomalidomide-induced ZMYM2 protein degradation in HEK293 and IMR32 cells, similarly as it is done on RNA level in Fig 4f. Effect sizes of protein degradation are rather small and only singlicate analyses are shown using WB as a readout. The study would benefit also from a global proteome analysis to monitor degradation of novel neo-substrates by quantitative mass spectrometry.

- Page 19, discussion: "Therefore, our results suggest that ZMYM2-FGFR1 degradation is one of the drug actions of pomalidomide in AML, caused by chromosomal translocation and the formation ZMYM2-FGFR1. In addition, our results indicate that ZMYM2-FGFR1 expression can be used to determine whether pomalidomide is a useful drug for a patient."

The authors need to describe here how frequent the ZMYM2-FGFR1 translocation occurs in AML. To this reviewer's knowledge it is very rare and only a handful of cases have been reported. Since pomalidomide is used in AML in a much wider area the statement here is misleading, because the ZMYM2-FGFR1 degradation mechanism of pomalidomide would only affect very few AML patients. The next claim that ZMYM2-FGFR1 expression can be used to determine drug efficacy of pomalidomide is also a very strong statement that is not supported by any data in the manuscript. This must be written in more hypothetical terms.

Minor points:

- Page 9: "These results indicate that drug-inducible biotinylation by AirID-CRBN with ThDs is fully agree with the selective interaction of three neo-substrates."

Rephrase...

- Page 11: Compare your ZNF results to Sievers et al, Science 2018

- Page 12: "These findings suggest that ZNF687 and ZMYM2 are novel neo-substrates of ThDs."

Small molecules have no substrates... rephrase

- Page 19: "As its protein degradation is less than the detectable level, the drug action of PROTAC is quite drastic."

Rephrase... Hard to understand

- Page 20: "Therefore, we believe that this method provides a powerful approach for understanding the overall IMiD-dependent interactome, including natural ligands, degradative proteins, and non-degradative proteins."

Regarding natural ligands the authors should further describe if they write about small molecule/metabolite natural ligands or peptides as natural ligands.

Point-by-Point Responses to the Reviewers' Critiques (NCOMMS-21-15078-T)

We deeply appreciate the thorough analysis and constructive suggestions provided by the three reviewers to guide us in improving our manuscript. As described in more detail below, we have addressed all the reviewers' concerns. With this extensive revision, we hope that the reviewers will agree that we have satisfactorily addressed all the concerns raised and have thereby substantially strengthened our paper.

Reviewer #1

This paper presents additional applications of the previously reported AirID-CRBN fusion to identify neosubstrates of CRBN with thalidomide and derivatives and PROTACs and extension to AirID-DCAF15 and the molecular glue indisulam. Although the AirID-CRBN system was briefly presented in a prior publication, showing that known neosubstrates are recruited to cereblon upon pomalidomide treatment [K. Kido et al., AirID, a novel proximity biotinylation enzyme, for analysis of protein-protein interactions. Elife 9, (2020)], this paper presents substantial new data, including discovery of a novel CRBN immunomodulatory drug-inducible neosubstrate with potential clinical significance, application to profiling PROTACs, and application to DCAF15. This paper would be a good fit for publication in Nature Communications with the following revisions addressed.

Response: We thank you for the considerate comments.

Reviewer suggestions:

- **Use of biotin site identification for quantification gives higher resolution, but is technically more challenging given the necessity of label free quantification and may be a barrier for use of this method by others. Do the authors see similar quantification values when evaluating these data by quantitative proteomics?**

Response: We thank you for pointing out a technical concern regarding the method of identifying the biotinylation site. In response to the reviewer's comment, we performed proteomic analyses using such conventional enrichment methods as streptavidin and anti-biotin antibody in MM1.S cells stably expressing AirID-CRBN. As a result, among the three enrichment methods for identifying the biotinylation site, the use of tamavidin 2-REV distinguished the most neo-substrates (Fig. 1d in the revised manuscript). Using tamavidin

2-REV also enabled us to identify the biotinylation sites of neo-substrates. Based on these results, we concluded that the method using tamavidin 2-REV is ideal for drug-inducible proteomics. In the revised manuscript, we have discussed the usefulness of the method involving tamavidin 2-REV (lines 133–141, 422–426).

• Presumably the identification of the novel zinc finger proteins in these cell lines by AirID-CRBN means that they are natively expressed in cells to a significant enough degree that the native protein would be able to be detected and validated by Western blot rather than expression of a fusion protein. A Western blot validating at least one or two of these targets in the appropriate cell line, particularly of the zinc finger of focus, ZMYM2, would support the authors conclusions about these proteins as novel targets.

Response: Thank you for your valuable and important suggestions regarding the protein expression levels of detectable biotinylated peptides using the AirID method. In response to the reviewer's comment, we compared the protein expression levels of ZNF687 and ZMYM2 in MM1.S, 293T, HuH7, and IMR32 cells. As a result, the protein expression levels of ZNF687 were almost the same in all cell lines (Supplementary Fig. 6c in the revised manuscript). By contrast, the protein expression level of ZMYM2 was the highest in IMR32 cells among all cell lines (Supplementary Fig. 6c in the revised manuscript). In the LC-MS/MS analyses in this study, biotinylated peptides of ZNF687 were detected in all cell lines while biotinylated peptides of ZMYM2 were detected only in IMR32 cells. Therefore, protein expression level is an important factor in determining whether this method detects biotinylated peptides of the protein. We added a new figure (Supplementary Fig. 6c) comparing the expression levels of ZNF687 and ZMYM2 and cited this in the revised manuscript (lines 205–211).

• In line with the above, the authors make an interesting but speculative connection of ZMYM2-FGFR1 to AML, but demonstration of this protein as a degraded target in an AML cell line would support. Pomalidomide does not have inherent activity against a number of AML cell lines, and statements in the discussion about the clinical relevance of these findings should be tempered.

Response: Thank you for pointing out the important and unresolved issue regarding the connection of ZMYM2-FGFR1 degradation to AML. Most recently, Ebert et al. reported that avadomide (CC-122) more strongly degrades ZMYM2 and ZMYM2-FGFR1 than pomalidomide (Renneville et al., *Blood Cancer Discov.* 2021;2(3):250-265). Furthermore, they showed that avadomide treatment reduced the number of viable cells in bone marrow CD34⁺ cells isolated from *ZMYM2-FGFR1*-positive hematologic malignancy patients (Renneville et al., *Blood Cancer Discov.* 2021;2(3):250-265). It is possible, therefore, that pomalidomide or avadomide is effective in the treatment of patients with *ZMYM2-FGFR1*-positive hematologic malignancy. While our study did not investigate the connection between ZMYM2-FGFR1 and AML, we discussed with hypothetical terms citing the work of Ebert in the revised manuscript (lines 445–450).

• **It is not clear why the non-PROTACs AlphaScreen data were performed at a single protein concentration. Optimization to justify the single concentration used should be included in the supplemental information.**

Response: Thank you for your important comment regarding the AlphaScreen-based biochemical assay using the wheat germ cell-free system. We previously showed that this assay system enables us to perform high-throughput assays of IMiD-dependent direct interactions without protein purification (Yamanaka et al., *EMBO J.* 2021;40(4):e105375 and Furihata et al., *Nat. Commun.* 2020;11(1):4578). Therefore—because we did not perform protein purification of the recombinant proteins—all biochemical assays in this study were performed at a single protein concentration. However, in the case of AlphaScreen using PROTAC, multiple concentrations of PROTAC were tried due to consideration of its “hook effect” (Supplementary Fig. 11d). In the case of IMiD-dependent interaction, we previously reported the interaction between CRBN and neo-substrates in a dose-dependent manner with IMiD (Yamanaka et al., *EMBO J.* 2021;40(4):e105375 and Furihata et al., *Nat. Commun.* 2020;11(1):4578), and the 20 μM of IMiDs used in this study (Fig. 4b) was a sufficient concentration in the previous study. In the revised manuscript, we explained why a single concentration of IMiD was used in this study (lines 245–248).

• **The authors state that “this method can identify degradation-independent interactors,” but this appears to be speculative with the current manuscript. The data**

indicate that this may be the case with ZNF536 and/or ZNF687, although additional validation would be necessary. Comparison of these zinc finger hits back to the consensus sequence that was identified in Ebert and co-workers, *Science*, 2018 would be helpful context for whether or not the AirID system is yielding an expected binding event or a novel binding event (as with ZMYM2).

Response: Thank you for your outstanding comment to strengthen our argument that this approach using AirID may enable us to identify non-degradative interactors. Ebert et al. mentioned that the 11 zinc fingers identified in their study did not have a consensus sequence (Sievers et al., *Science* 2018;362(6414):eaat0572). Therefore, based on further analyses, they concluded that the 11 zinc fingers had a low degree of sequence similarity, and that only distinct amino acid combinations at the drug–CRBN interface resulted in protein degradation (Sievers et al., *Science* 2018;362(6414):eaat0572). On the other hand, they mentioned that docking analysis based on crystal structure information can predict the interaction of C2H2-ZNFs with CRBN-IMiDs. In response to the reviewer’s comment, we confirmed the score of the zinc fingers in the ZNF536 or ZNF687 *in silico* analysis data (Sievers et al., *Science* 2018;362(6414):eaat0572). As a result, all zinc fingers in ZNF536 and ZNF687 showed low ranking in their *in silico* analyses. However, our AlphaScreen-based biochemical analyses showed that ZNF536 and ZNF687 interact directly with CRBN in the presence of IMiD. Therefore, we concluded that this method can identify degradation-independent interactors because it does not depend on protein degradation, although many analyses will be required to reveal how such degradation-independent interactors function in IMiD responses in the future. In the revised manuscript, we mentioned the consensus sequence among zinc fingers by citing Ebert’s work in the Discussion section (lines 451–467).

• Figure 1C. WT and YW/AA CRBN expression are very uneven and may be the underlying reason for the difference in biotinylation of SALL4.

Response: We thank you for pointing out an important concern regarding the influence on biotinylated SALL4 caused by different protein expression levels between AirID-CRBN-WT and AirID-CRBN-YW/AA. In response to the reviewer’s comment, we

performed a streptavidin pull-down assay using HEK293T-CRBN^{-/-} cells overexpressing SALL4 and CRBN-WT or -YW/AA, and the results have been added to Supplementary Fig. 1b. As shown in this revised manuscript, AirID-CRBN-WT pomalidomide-dependently increased biotinylation of SALL4, but AirID-CRBN-YW/AA—with the same expression level as AirID-CRBN-WT—did not (Supplementary Fig. 1b in the revised manuscript). This result indicates that the biotinylation of SALL4 is induced by the interaction between CRBN and pomalidomide.

- **Figure 1E and others, there are blots (particularly tubulin enriched) that are so light it is unclear if anything is shown. Please put a border around the blots or adjust the greyscale so the contrast is clearer.**

Response: We have put a border around all blots in the revised manuscript.

- **Several suggestions for clarification of the text in the manuscript are provided below:**

- **“by infection of lentivirus with HEK293T-CRBN^{-/-} cells”, please rephrase to “by infection of HEK293T-CRBN^{-/-} cells with lentivirus” (89-90)**

Response: We have corrected the text in the revised manuscript (lines 92–93).

- **“which is an unbound form with CRBN” (273), please rephrase to “which is a form unbound by CRBN”**

Response: We have rephrased the text in the revised manuscript (line 349).

- **“lenalidomide and pomalidomide are the main drugs for treating hematologic cancer, including multiple myeloma (MM)” (38-39) should say some types of hematologic cancer**

Response: We have corrected the text in the revised manuscript (line 39).

- **“A small molecule in cells can reportedly regulate biological functions via the induction of PPIs” (47-48) Please remove “reportedly”**

Response: We have removed “reportedly” in the revised manuscript (line 47).

- **“Indisulam is a second molecular glue in mammalian cells” (285) – indisulam defines a second class of immunomodulatory molecular glues, but is not a second molecular glue.**

Response: We have corrected the text regarding indisulam in the revised manuscript (lines 360–361).

- **“Immunomodulatory drugs (IMiDs) such as thalidomide, lenalidomide, and pomalidomide have been the most characterised molecular glues” (331-332) – this is arguable relative to molecular glues like rapamycin and cyclosporin.**

Response: We have corrected the text in the revised manuscript (lines 409–411).

- **“it is possible that there are unknown neo-substrates containing other types of ZNFs, although many analyses are required to identify such neo-substrates (354-355) – seems as though using this technique will mitigate these challenges.**

Response: We have rephrased the text in the revised manuscript (lines 436–437).

- **“bitotinylated” (63) – typo**

Response: We have corrected the mistake in the revised manuscript (line 62).

- **Kd ~ 10-15 (81) - needs units**

Response: We have corrected the mistake in the revised manuscript (line 82).

- “the full length form” (143) - of what?

Response: We have corrected the mistake in the revised manuscript (lines 162–163).

- “however, these phosphorylation levels were almost the same in both cell lines” (277-278) with or without drug treatment or both?

Response: We have removed this sentence in the revised manuscript (lines 350–352).

- Unclear what “high drug action” means (used several times in PROTAC section)

Response: We have clarified the meaning of “high drug action” in the revised manuscript.

Reviewer #2

In their manuscript, Yamanaka et al. performed comprehensive profiling of ubiquitin ligase interactors and drug-induced neo-substrates using proximity biotinylation. To this end, the newly developed ancestral BirA (AirID) which was described by the group in a previous publication (Kido et al., *elife* 2020) was fused to CRBN and other ubiquitin ligases, expressed in various cell lines, treated with drugs and subsequently biotin precipitation followed by mass spectrometry was performed to identify proteins in proximity of the ubiquitin ligase. Furthermore, the authors demonstrate that AirID technology can be successfully extrapolated to analyze PROTAC-based protein interactions. These analyses identified the major known neo-substrates for molecular glues including IKZF1/3, CK1A for lenalidomide, RBM39 for indisulam, and BRD4 for a BRD4-PROTAC, demonstrating the applicability and great potential of this approach that can overcome the low sensitivity of conventional co-immunoprecipitation that is hampered by weak enzyme-substrate interaction. ZMYM2, a potential target in cancer, was identified as a novel neo-substrate for pomalidomide/CRBN and subsequent analyses found that ZMYM2 interacts via its MYM-type zinc finger domain in contrast to C2H2-ZNF present in other thalidomide neo-substrates. The experiments are very well performed and of high technical quality.

The work provides a systematic and comprehensive overview of drug-dependent interactions of CRBN and other ligases. However, the AirID method has been described previously including pomalidomide/CRBN as an example by the group (Kido et al., *elife* 2020), and most of the neo-substrates/ interactors were described before in several publications. So to me the major novel biological finding of this work is the identification of ZMYM2 as a pomalidomide-specific substrate that may be translated to a clinical application in cancer but unfortunately this is not further followed up in the current paper. In a very recent publication in *Blood Cancer Discovery* by Renneville et al. (<https://bloodcancerdiscov.aacrjournals.org/content/2/3/250>, not yet on Pubmed) identified ZMYM2 as a novel avadomide-dependent CRBN substrate. They demonstrated that avadomide is more potent than pomalidomide in degrading ZMYM2. This finding was corroborated with in vitro and in vivo analysis including human patient samples. Although this supports the use of ZMYM2 as target in cancer, it also takes some of the novelty of the current work.

Response: We thank you for the considerate comments and important concerns regarding the novelty of this study.

Specific remarks:

- Besides drug-induced neo-substrates, does AirID also detect other interacting proteins like E3 ligase complex members (DDB1, CUL4 etc.) or E2s in the absence of drugs? This is also of interest to the field and should be reported.

Response: Thank you for your valuable comment about the proteins interacting with CRBN in the absence of the drug. In response to the reviewer's comment, we confirmed whether biotinylated peptides of drug-independent CRBN interactors—such as CRL4 complex members or E2 ubiquitin-conjugating enzymes—were detected. As a result, DDB1, CUL4A and CUL4B—which are CRL4 complex members—were detected by LC-MS/MS analyses. In addition, UBE2M, which is a NEDD8-conjugating enzyme of CRL4^{CRBN} (Pan et al., *Oncogene* 2004;23:1985-1997), was detected in the LC-MS/MS analyses. On the other hand, UBE2D3 and UBE2G1, which are E2 ubiquitin-conjugating enzymes for protein degradation of neo-substrates by CRL4^{CRBN} (Lu et al., *eLife*

2018;7:e40958), were not detected in our LC-MS/MS analyses. In addition, COP9 signalosome components, which are involved in de-neddylation of the CRL complex (Olma et al., *J cell Sci.* 2009;122(7):1035-1044, Fischer et al., *Cell* 2011;147:1024-1039), were detected in the LC-MS/MS analyses. These results indicate that this method also enables the identification of CRL4^{CRBN} interactors. We added a new table regarding the CRBN interactors to Supplementary Fig. 7a and discussed the CRBN interactors in the revised manuscript (lines 214–217).

• **There are several described natural substrates of CRBN like MEIS2, MCT-1, CD98, and GL2 that in part are also influenced by thalidomide analogs. Have these proteins been detected in treated/ untreated cells?**

Response: Thank you for making the valuable comment about the natural substrates of CRBN. In response to the reviewer's comment, we examined whether biotinylated peptides of natural substrates of CRBN were detected by LC-MS/MS analyses. Biotinylated peptides of TP53 (Zhou et al., *Cell Death Dis.* 2019;10(2):69), p97/VCP (Nguyen et al., *PNAS* 2017;114(14):3565-3571), SQSTM1/p62 (Zhou et al., *Hum Mol Genet.* 2018;24(4):667-678) and HSP90 (Heider et al., *Mol Cell.* 2021;81(6):1170-1186), which are natural substrates of CRL4^{CRBN}, were detected in the LC-MS/MS analyses. However, biotinylation levels of some interacting proteins were increased or decreased, but there were no significant changes as in the neo-substrates. On the other hand, MEIS2 (Fischer et al., *Nature* 2014;512(7512):48-53), MCT-1 (Eichner et al., *Nat. Med.* 2016;22(7):735-743), CD98 (Heider et al., *Mol Cell.* 2021;81(6):1170-1186) or GS (Nguyen et al., *Mol Cell.* 2016;61(6):809-820) were not detected in the LC-MS/MS analyses. Based on these results, we concluded that cell-lines and culture conditions may be critical for analysing natural substrates by IMiD supplementation. We added a new table regarding the CRBN interactors to Supplementary Fig. 7b and discussed the CRBN interactors in the revised manuscript (lines 217–223).

• **Is there a correlation between the degree of biotinylation of a neo-substrate and its ubiquitination/degradation?**

Response: We thank the reviewer for raising an important question about the correlation between biotinylation of a neo-substrate and its protein degradation. To address these concerns experimentally, we investigated the degree of protein degradation of neo-substrates by AirID-CRBN in a dose-dependent manner. In HEK293T-CRBN-KO cells stably expressing AirID-CRBN, 8 h of treatment with pomalidomide scarcely induced protein degradation of ZFP91 and PLZF (Supplementary Fig. 1g). By contrast, in the case of HuH7 cells stably expressing AirID-CRBN, endogenous CRBN, ZFP91, PLZF, and SALL4 degradation occurred after 8 h of treatment with pomalidomide (Supplementary Fig. 1h). Because AirID-CRBN forms the CRL4^{CRBN} complex as described in the above response, it was indicated that IMiD-dependent protein degradation is weakened by fusion with AirID. Based on these results, we concluded that AirID-fused CRBN interacts with the neo-substrate in the CRL4 complex, but the degradation activity was weaker. We have added additional experimental results to Supplementary Fig. 1g, h and discussed the correlation between biotinylation and degradation in the revised manuscript (lines 114–122).

• Neo-substrate biotinylation differs between cell lines. Could this be due to different levels of expression of the BirA-CRBN in these cells? What other reasons could explain this?

Response: Thank you for your outstanding query regarding the differences in biotinylated peptides between cells. We agree with the reviewer's comment that the different expression levels of AirID-CRBN are one of the reasons for the differences in biotinylated peptides. In addition, we believe that differences in the profile of protein expression and its expression level are the most influential reasons for the differences in detected biotinylated peptides. As shown in Supplementary Fig. 6b, more biotinylated peptides were detected in IMR32 cells than in HEK293T cells, although the expression levels of AirID-CRBN were almost the same in the two cell lines. In addition, to address this concern experimentally, we compared the expression levels of several ZNF clones in MM1.S, HEK293T, HuH7, and IMR32 cells. As a result, the protein expression levels of ZNF687 were almost the same in all cell lines (Supplementary Fig. 6c in the revised manuscript). By contrast, the protein expression level of ZMYM2 was the highest in IMR32 cells among all cell lines (Supplementary Fig. 6c in the revised manuscript). In the LC-MS/MS analyses in this study,

biotinylated peptides of ZNF687 were detected in all cell lines and biotinylated peptides of ZMYM2 were detected in IMR32 cells. We therefore concluded that the profile of protein expression in each cell line—rather than the AirID-CRBN expression level—is the critical reason for differences in neo-substrate biotinylation. We added a new figure to Supplementary Fig. 6c and mentioned the differences in biotinylated proteins in the revised manuscript (lines 199–211).

• Several ZN finger proteins like ZNF536, 687, and GSPT1 were found biotinylated by Airid-CRBN in the presence of lenalidomide and pomalidomide but no or only weak degradation is observed. How is this explained?

Response: We thank the reviewer for the important comment regarding the correlation between interaction and degradation. Thalidomide and pomalidomide cannot induce CK1 α degradation, although the two compounds induce an interaction between CRBN and CK1 α , as previously reported (Petzold et al., *Nature* 2016;532(7597):127-130). In addition, Thomä et al. concluded that these differences in binding affinity are the most critical factor for protein degradation of neo-substrates (Sievers et al., *Science* 2018;362(6414):eaat0572). On the other hand, they revealed that CC-122 and CC-220 induced protein degradation of C2H2-ZNFs that were not degraded by pomalidomide (Sievers et al., *Science* 2018;362(6414):eaat0572). Therefore, it is possible that ZNF536 and ZNF687 are degraded by other thalidomide derivatives such as GSPT1–CC-885 (Matyskiela et al., *Nature* 2016;535(7611):252-257). However, to understand whether ZNF536 and ZNF687 are degraded by other thalidomide derivatives or are IMiD-dependent interacting proteins, but not neo-substrate, further analyses are required in future research. In the revised manuscript, we have discussed non-degradative interacting proteins (lines 451–467).

• P. 7, line 150; see Petzold et al., 2016 for detailed measurements of CK1a-CRBN binding in the presence of other IMiDs, CK1A binding in vitro is also observed with thalidomide, lenalidomide and pomalidomide and small differences in binding Kd translate into almost binary degradation/no-degradation choices in cells; the Matyskiela et al., 2016 reference refers to GSPT1; this should be corrected and PMID: 26909574 should be cited instead.

Response: Thank you for pointing out an important concern regarding the neo-substrate specificity of IMiD, as revealed in previous reports. In response to the reviewer's comment, we corrected the details on CK1 α protein degradation and interaction by referring to the correct reference (Petzold et al., *Nature* 2016;532(7597):127-130) in the revised manuscript (line 179).

• **The text and language needs to be substantially revised. Some sentences are illogical or hard to understand and there are many grammatical errors.**

Example:

“In the case of the pan-ThD neo-substrate PLZF11, biotin supplementation increased its biotinylation in HEK293T cells (Extended Data Fig. 1c). However, biotinylated PLZF was not detected upon supplementation with the highest biotin concentration (500 μ M).”

Response: We thank you for the important suggestion regarding the quality of the manuscript. We have thoroughly revised the manuscript with the assistance of a native English-speaking science editor at Editage (<https://www.editage.com>).

• **ThD is not a commonly used abbreviation, I would either spell out or use IMiD for immunomodulatory drug**

Response: We have rephrased ThD to IMiD in the revised manuscript.

Reviewer #3

In their study Yamanaka et al. claim to have developed a new method for drug-inducible protein proximity analysis using a previously reported (Kido et al, eLIFE 2020) further developed version of the BioID biotin ligase, named AirID. The authors first introduce the method that allows to identify proteins in proximity to molecular glues (here IMiDs, indisulam) or PROTAC drugs (dBET1/ARV-825 for BRD4) by ectopic expression of a fusion protein of a drug-binding E3 ligase (CRBN, DCAF15) and the AirID biotin ligase for proximity labeling. The method robustly works in living cells across multiple cell lines and allows to identify well known neo-substrates of lenalidomide-bound CRBN, such as IKZF1 or IKZF3, with biotin

WB and mass spectrometry as a read out. Of particular interest are the identification of ZMYM2 and its FGFR1 fusion protein ZMYM2-FGFR1 as novel neo-substrates of CRBN, as demonstrated by AirID proximity labeling, Alpha-Screen binding assays and neo-substrate degradation assays in living cells. ZMYM2-FGFR1 is an oncogene in very rare cases of acute myeloid leukemia (AML). Consequences on oncogenic signal transduction after ZMYM2-FGFR1 degradation via pomalidomide are demonstrated in HEK293 cells, but not in an AML relevant model, patient cells or an animal model.

The study is technically sound and the manuscript is mostly well written.

Response: We thank you for the considerate comments.

Major points of criticism:

- The title is misleading – “Drug-inducible protein proximity analysis using AirID” – sounds like a small molecule activates a BioID enzyme, which is not the case. The small molecule rather changes the affinity of the E3 ligase for neo-substrates

Response: Thank you for your valuable suggestion regarding the title of this study. In response to the reviewer’s comment, we have corrected the article title to “Molecular glue- and PROTAC-inducible biotinylation analysis using proximity biotinylation enzyme AirID.”

- The authors claim that their drug-induced BioID approach for molecular glues is a new technical approach. The methodological advancement is incremental though as compared to their earlier report (Kido et al, eLIFE 2020) and a very similar approach has been published previously by Pech et al (eLIFE 2019; <https://pubmed.ncbi.nlm.nih.gov/31452512/>). Unfortunately, the authors do not at all cite this study, although Pech et al have used a BioID approach to confirm RBM39-degradation via DCAF15 after indisulam treatment: “DCAF15-BioID cells were pre-treated with indisulam for 72 hr prior to biotin and MG132 addition. This treatment regime lead to substantial indisulam-dependent biotinylation of RBM39”. This is very similar to the author’s Figure 6 c in Yamanaka et al.

Response: Thank you for pointing out the important concerns about previous reports using cells expressing DCAF15-BioID. As shown in our previous report (Kido et al., *eLife* 2020;11(9):e54983), AirID has higher proximity-dependent biotinylation activity than BioID and enables biotinylation of interacting proteins in shorter reaction times (Kido et al., *eLife* 2020;11(9):e54983). In a study by Pech, pre-treatment with indisulam was performed for 72 h prior to biotin and MG132 supplementation (Pech et al., *eLife* 2019;27(8):e47362). Given that small molecule-dependent protein degradation of neo-substrates may affect cell viability, such as IMiD-Ikaros/Aiolos and Indisulam-RBM39 in specific cell lines, shorter biotinylation time is an important advantage. In the revised manuscript, we discuss the differences between the AirID method and the BioID approach by citing Pech's report in the Results section (lines 275–281, 362–365).

• **Page 2, abstract:** “Here, we demonstrate a new approach using AirID, a proximity biotinylation enzyme, to analyse target proteins proximately interacting with drugs in cells, such as molecular glues and PROTACs.” The approach is not entirely new, see Pech et al, *eLIFE* 2019.

Response: Thank you for your important concern about the novelty of drug-dependent protein proximity using a proximity biotinylation enzyme. In the revised manuscript, we mentioned the novelty of the method using AirID and tamavidin 2-REV rather than the use of proximity-dependent biotinylation enzyme (line 24). As described above, we have corrected the context by citing Pech's report in the revised manuscript (line 275–281, 362–365).

• **From a technology standpoint it is a major limitation of this and also the original AirID study, that the AirID biotin ligase is not properly benchmarked against the widely used TurboID (and also BirA*) biotin ligase in proximity proteomics experiments in living cells. The authors obviously propagate their AirID version of the BioID enzyme, but is it really better than TurboID for assays in living cells? The authors should show a proper comparison in living cells at low biotin concentrations and short labeling times for AirID vs TurboID with quantitative mass spectrometry as a readout (use pomalidomide/CRBN in IMR32). Hundreds of laboratories use BirA* or TurboID, why switch to AirID?**

Response: Thank you for pointing out an important unresolved concern regarding benchmarks among AirID, TurboID, and BioID (BirA*). To address this experimentally, we generated HEK293T and IMR32 cells stably expressing AirID-CRBN, TurboID-CRBN, or BioID-CRBN using the same procedure. First, we confirmed IMiD-dependent biotinylation of neo-substrates by immunoblot analysis after the streptavidin pull-down assay. BioID-CRBN was unable to biotinylate neo-substrates under the same conditions as AirID-CRBN. Furthermore, AirID-CRBN IMiD-dependently biotinylated neo-substrates under both short (2 h) and long (6 h) labelling conditions. On the other hand, TurboID-CRBN also biotinylated neo-substrate in the presence of pomalidomide under short labelling conditions (2 h), but the IMiD-dependency of TurboID-CRBN was lower than that of AirID-CRBN under long labelling conditions (6 h). Accordingly, we performed LC-MS/MS analyses using IMR32 cells expressing AirID-CRBN or TurboID-CRBN under the same biotin concentration (10 μ M) and short labelling time (2 h). As a result, AirID-CRBN and TurboID-CRBN biotinylated neo-substrates, such as SALL4 and WIZ, in the presence of pomalidomide (Fig. 5e in the revised manuscript). On the other hand, some neo-substrates were biotinylated only in AirID-CRBN or Turbo-ID expressing IMR32 cells (Fig. 5e in the revised manuscript). Furthermore, biotinylated proteins differed between AirID-CRBN and TurboID-CRBN (Fig. 5f in the revised manuscript). Based on these results, we concluded that AirID-CRBN and TurboID-CRBN have distinct properties, and both enzymes are useful for drug-dependent proximity analysis under brief labelling conditions. In the revised manuscript, we added the additional experimental results to Fig. 5 and discussed the benchmark among the three biotinylation enzymes in the revised manuscript (line 270–305).

• With respect to AirID proximity proteomics it is also unclear why the authors only use the streptavidin 2-REV enrichment approach. Alternatives are capture of biotinylated peptides by widely available antibodies (as in Udeshi et al, Nat. Methods 2017; <https://pubmed.ncbi.nlm.nih.gov/29039416/>) or streptavidin pull downs with harsh washing conditions and on bead digest, which provide a lot better sensitivity but less selectivity as compared to the biotinylated peptide captures. The study would benefit a lot from a technological perspective if these approaches were compared in an application for identifying neo-substrates of CRBN.

Response: Thank you for raising an important unresolved concern regarding the comparison of enrichment methods. To address this concern, we performed enrichment of biotinylated peptides using tamavidin 2-REV or anti-biotin antibody followed by LC-MS/MS. We also performed enrichment of biotinylated proteins using streptavidin, followed by bead digestion and LC-MS/MS (Fig. 1d in the revised manuscript). As a result, tamavidin 2-REV enrichment enabled us to detect the most neo-substrates among the three approaches. Furthermore, the method using tamavidin 2-REV enabled us to identify the biotinylation sites of the neo-substrate. Therefore, we concluded that the method using tamavidin 2-REV is ideal for drug-inducible proteomics. In the revised manuscript, we added the results to Fig. 1d and discussed the comparison among the enrichment methods (lines 133–141, 422–426).

• **Figure 4e: Quantify effect of pomalidomide-induced ZMYM2 protein degradation in HEK293 and IMR32 cells, similarly as it is done on RNA level in Fig 4f. Effect sizes of protein degradation are rather small and only singlicate analyses are shown using WB as a readout. The study would benefit also from a global proteome analysis to monitor degradation of novel neo-substrates by quantitative mass spectrometry.**

Response: Thank you for pointing out an important concern regarding the quantitative analyses of neo-substrates. To address this comment, we performed quantitative analysis of ZMYM2 degradation by fluorescent immunoblotting and added the results to the revised manuscript (Fig. 4g in the revised manuscript). As a result, ZMYM2 was significantly degraded in the presence of pomalidomide in IMR32 cells (Fig. 4g in the revised manuscript). Furthermore, in a very recent study, it was reported that pomalidomide and avadomide induced protein degradation of ZMYM2 by quantitative mass spectrometry (Renneville et al., *Blood Cancer Discov.* 2021;2(3):250-265). In the revised manuscript, we added the results of the quantitative immunoblot analysis to Fig. 4g in the revised manuscript.

• **Page 19, discussion: “Therefore, our results suggest that ZMYM2-FGFR1 degradation is one of the drug actions of pomalidomide in AML, caused by chromosomal translocation and the formation ZMYM2-FGFR1. In addition, our**

results indicate that ZMYM2-FGFR1 expression can be used to determine whether pomalidomide is a useful drug for a patient.” The authors need to describe here how frequent the ZMYM2-FGFR1 translocation occurs in AML. To this reviewer’s knowledge it is very rare and only a handful of cases have been reported. Since pomalidomide is used in AML in a much wider area the statement here is misleading, because the ZMYM2-FGFR1 degradation mechanism of pomalidomide would only affect very few AML patients. The next claim that ZMYM2-FGFR1 expression can be used to determine drug efficacy of pomalidomide is also a very strong statement that is not supported by any data in the manuscript. This must be written in more hypothetical terms.

Response: Thank you for your important unresolved concern about the correlation between ZMYM2-FGFR1 degradation and pomalidomide treatment for AML. We also understand that AML caused by ZMYM2-FGFR1 translocation is very rare. Therefore, we have mentioned this frequency in the revised manuscript (lines 439–440). Regarding the efficiency of treatment of AML caused by ZMYM2-FGFR1 translocation, Ebert et al. reported that avadomide (CC-122) degraded ZMYM2-FGFR1 more strongly than did pomalidomide, and showed that avadomide treatment reduced the number of viable cells among bone marrow CD34⁺ cells isolated from ZMYM2-FGFR1-positive hematologic malignancy patients (Renneville et al., *Blood Cancer Discov.* 2021;2(3):250-265). Therefore, it is possible that pomalidomide or avadomide is effective in the treatment of patients with ZMYM2-FGFR1-positive hematologic malignancy. In our study—because we did not investigate the connection between ZMYM2-FGFR1 and AML—we discussed with hypothetical terms citing the work of Ebert in the revised manuscript (lines 445–450).

Minor points:

- Page 9: “These results indicate that drug-inducible biotinylation by AirID-CRBN with ThDs is fully agree with the selective interaction of three neo-substrates.”

Rephrase...

Response: We have rephrased the sentence in the revised manuscript (lines 180–181).

- Page 11: Compare your ZNF results to Sievers et al, Science 2018

Response: We compared our ZNF results to Sievers et al, Science 2018, and mentioned the alignments in the revised manuscript (lines 451–467).

• **Page 12:** “These findings suggest that ZNF687 and ZMYM2 are novel neo-substrates of ThDs.”

Small molecules have no substrates... rephrase

Response: We have rephrased the sentence in the revised manuscript (lines 259–260).

• **Page 19:** “As its protein degradation is less than the detectable level, the drug action of PROTAC is quite drastic.”

Rephrase... Hard to understand

Response: We have rephrased the sentence in the revised manuscript (lines 469–470).

• **Page 20:** “Therefore, we believe that this method provides a powerful approach for understanding the overall IMiD-dependent interactome, including natural ligands, degradative proteins, and non-degradative proteins.”

Regarding natural ligands the authors should further describe if they write about small molecule/metabolite natural ligands or peptides as natural ligands.

Response: We have clarified this in the revised manuscript (lines 499–501).

REVIEWERS' COMMENTS

Reviewer #1 (Remarks to the Author):

The authors have substantially revised the manuscript to address all reviewer concerns (myself and other reviewers). The approach is now benchmarked across cell types, compounds, enrichment methods, and different proximity labeling fusion proteins. The findings are robust and will be of interest to many seeking to identify neosubstrates of these and similar compounds. I enjoyed reading this revision and look forward to seeing the paper in print.

Reviewer #2 (Remarks to the Author):

The authors have substantially revised their manuscript and addressed almost all my concerns. The new data is very helpful to appraise their new biotinylation-based method for examining the interaction space of ubiquitin ligases in the context of different protein degraders. Since this manuscript describes a new applicable method, making the AirID plasmids easily available (e.g. Addgene) would further enhance its impact.

I just have some minor comments:

The new title “Molecular glue and PROTAC-inducible biotinylation analysis using proximity biotinylation enzyme AirID.” is still not very clear. In the study biotinylation analysis with AirID is used to identify molecular glue and PROTAC-induced protein interaction.

The text needs again to be carefully revised:

Some examples:

line 172: “using the optimal conditions above,...” should be “using the optimal conditions described above,...”

Line 221: the term “ubiquitination” would be more appropriate than “proteasomal degradation” for CRL4-CRBN E3 ligase function

Line 481: “PROTACs are the most attractive technology for inducing degradation of target proteins...”, in regard to the clinical success and better pharmacologic properties of molecular glues I would not call PROTACs the most attractive but rather another or alternative technology.

Reviewer #3 (Remarks to the Author):

The authors have successfully addressed all of my concerns and have further improved their manuscript. Please note that the line numbers in the response letter did not match the line numbers in the manuscript.

Point-by-Point Responses to the Reviewers' Critiques (NCOMMS-21-15078A)

We deeply appreciate the constructive suggestions and kind comment to publish this study. As described in more detail below, we have addressed all the reviewers' concerns.

Reviewer #1

The authors have substantially revised the manuscript to address all reviewer concerns (myself and other reviewers). The approach is now benchmarked across cell types, compounds, enrichment methods, and different proximity labeling fusion proteins. The findings are robust and will be of interest to many seeking to identify neosubstrates of these and similar compounds. I enjoyed reading this revision and look forward to seeing the paper in print.

Response: We thank you for the kind comments on our revised manuscript.

Reviewer #2

The authors have substantially revised their manuscript and addressed almost all my concerns. The new data is very helpful to appraise their new biotinylation-based method for examining the interaction space of ubiquitin ligases in the context of different protein degraders. Since this manuscript describes a new applicable method, making the AirID plasmids easily available (e.g. Addgene) would further enhance its impact.

Response: We thank you for the kind comments on our revised manuscript. Following the reviewer's suggestion, we plan to make the AirID plasmid available from Addgene.

I just have some minor comments:

The new title “Molecular glue and PROTAC-inducible biotinylation analysis using proximity biotinylation enzyme AirID.” is still not very clear. In the study biotinylation analysis with AirID is used to identify molecular glue and PROTAC-induced protein interaction.

Response: Thank you for your valuable suggestion regarding the title of this study. In response to the reviewer's comment, we have corrected the article title to “A proximity biotinylation-based approach to identify protein-E3 ligase interactions induced by PROTACs and molecular glues.”

The text needs again to be carefully revised:

Some examples:

line 172: “using the optimal conditions above,...” should be “using the optimal conditions described above,...”

Response: We have corrected the mistake in the revised manuscript (line 165).

Line 221: the term “ubiquitination” would be more appropriate than “proteasomal degradation” for CRL4-CRBN E3 ligase function

Response: We have rephrased the text in the revised manuscript (line 213).

Line 481: “PROTACs are the most attractive technology for inducing degradation of target proteins...”, in regard to the clinical success and better pharmacologic properties of molecular glues I would not call PROTACs the most attractive but rather another or alternative technology.

Response: We have corrected the sentence in the revised manuscript (line 468).

Reviewer #3

The authors have successfully addressed all of my concerns and have further improved their manuscript. Please note that the line numbers in the response letter did not match the line numbers in the manuscript.

Response: We thank you for the kind comments on our revised manuscript. We also apologize for the mistake in the line number between response letter and the revised manuscript.